# AN INFORMATION-THEORETIC PARAMETER-FREE BAYESIAN FRAMEWORK FOR PROBING LABELED DEPENDENCY TREES FROM ATTENTION SCORE

**Hongxu Liu**[1]   **Jing Ma**[2]   **Xiaojie Wang**[2] *   **Caixia Yuan**[2]   **Fangxiang Feng**[2]
[1]Nanyang Technological University, Singapore
[2]Beijing University of Posts and Telecommunications, Beijing, China
`hongxu001@e.ntu.edu.sg`   `{majing,xjwang,yuancx,fxfeng}@bupt.edu.cn`

## ABSTRACT

Figuring out how neural language models comprehend syntax acts as a key to revealing how they understand languages. We systematically analyzed methods for finding syntax structures in models, namely *probing*, and found limitations yet widely exist in previous probing practice. We proposed a method capable of estimating mutual information (MI) and extracting dependency trees from attention scores in a mathematical-rigorous way, requiring no additional network training effort. Compared with previous approaches, it has a much simpler model, while being able to probe more complex dependency trees, also transparent for fine-grained explanation. We tested our method on several open-source LLMs and demonstrated its effectiveness by systematically comparing it with a great many competitive baselines. Several informative conclusions can be drawn by further analysis of the results, shedding light on our method's explanatory potential. Our code is released at `https://github.com/ChristLBUPT/IPBP`.

## 1 INTRODUCTION

Recent advancements in Large Language Models (LLMs) have left the world with deep impressions. This process is accompanied by confusion, since LLMs are largely black-box and usually trained on simple next-token-prediction LM tasks. While the interpretability community recently assigns great importance to mechanistic (circuit-based) model explaining (Elhage et al., 2021; Wang et al., 2022; Ferrando & Voita, 2024), traditional *probing* methods, which aim at extracting syntax structures from model states, are still worth working on for two reasons: 1. They can provide dataset-wide conclusions, while most circuit-based methods tend to be sample-wise (except for Elhage et al. (2021), which is a static parameter analysis) 2. Syntax structures are among the most topologically-complicated and all-around concepts about languages. They are also essential to humans' language comprehension, verified by various brain studies (Lopopolo et al., 2021; Dotan & Brutmann, 2022; Fallon & Pylkkänen, 2024).

A common practice of syntactical probing is to train a supervised classifier network on top of model states (Hewitt & Manning, 2019; Pimentel et al., 2020b; Müller-Eberstein et al., 2022) to predict dependency syntax trees, or directly take some model states as evidence for syntax (Htut et al., 2019). Despite the insights they gave, it is obvious that most of the previous probing methods are **explaining by unexplainibility**, meaning that most of them are introducing external trainable networks to extract syntax, ranging from simple linear mappings (Liu et al., 2019) to deep MLPs (Hewitt & Liang, 2019; Voita & Titov, 2020; Pimentel et al., 2020b) or pseudo attention heads (Pimentel et al., 2022). This results in a trade-off: Linear mappings are simple and explainable, but have limited expressivity. Deeper networks can fit any co-relationships, but a deep probing network itself is hard to explain, so it's natural to raise the doubt on whether the extracted syntax structures really come from the probed LM, or just the strong probes have learned to unconditionally predict them. This gets even worse for modern LLMs that have larger hidden dimensions compared with pre-trained models, making trainable probing networks inevitably larger to fit in the dimensionality.

---

*Corresponding Author

If we dive deeper, we might find some clues about this bitter tradeoff: previous methods put their attention mainly on contextualized hidden states. Since hidden state vectors have rather different structures compared with syntax trees, a trainable mapping network is necessary. Using hidden states is also a primary cause for the aforementioned concern of *the probe itself learns the task*: Contextualized embeddings embed abundant semantics, so an extreme case is that even if the probed LM knows nothing about syntactics, it's still possible that the deep probing model learns it (see Hewitt & Liang (2019)). A simple thought experiment is that, given two single words *eat* and *breakfast*, it is plausible that *breakfast* acts as the noun object of the verb *eat* even without giving any context. This is exactly the case of this concern.

If hidden states aren't good enough, what's the better choice? Maybe we should *put attention on attention*. Attention is the only component that involves inter-token relationships (MLP and add/norm are applied token-wise), while dependency syntax is also an inter-word relationship. Attention scores are stacked matrices, while dependency trees can also be described as adjacency matrices (See Section 3.1 for details). This means that attention maps are conceptually, also topologically consistent with dependency syntax trees.

Unfortunately, despite those nice consistencies, there are only few methods focusing on probing attention for syntax (Clark et al., 2019; Htut et al., 2019; Vig & Belinkov, 2019; Ravishankar et al., 2021), only being able to extract low-quality or incomplete dependency trees. A seemingly good choice yields results that are not desirable. Why is it? This introduces the second limitation of previous methods, **over-trusting attention scores**: Due to the aforementioned nice consistencies, together with the fact that (softmax-normalized) attention scores form valid probability distributions across tokens, so it seems natural to directly take them as probability distributions of the dependency relationships of words. However, we must admit that attention scores wouldn't need to be dedicated to syntax, meaning that filtering out highly syntactical attention heads and explainable transformations on raw scores are necessary.

Based on our analysis, we proposed a method called **I**nformation-theoretic **P**arameter-free **B**ayesian **P**robing (IPBP): Instead of training supervised networks on hidden states, we chose to directly estimate the multivariate distributions between attention scores and dependency relationships, and integrate on those distributions to calculate mutual information (MI) between each attention head's scores and the existence of a certain dependency relationship in a mathematically closed form. Since it's parameter-free and attention-based, it almost denies the possibilities of *the probe itself learns the task*, as mentioned before. We further designed a well-formed decoding algorithm incorporating the estimated MI and Bayesian posteriors, being able to efficiently reconstruct *labeled* dependency trees, while preventing us from dropping into the trap of using attention scores themselves as dependency probabilities. Comparisons with a series of strong baselines indicate that our method has the best head importance estimation and tree-constructing qualities. We further derived informative conclusions on the estimated MI and distributions, including the one obsessing probing researchers for a long time: *do model layers accord with tree heights*? (A similar question once discussed by Tenney et al. (2019a)). In a word, our method addresses the two limitations in an elegant way, while offering vast possibilities for the upcoming conclusion-intensive research thanks to its fine-grained MI and probability functions.

## 2 RELATED WORK

Just after the birth of deep contextualized embeddings (Peters et al., 2018) and transformer-based pre-trained models (Devlin et al., 2019), researchers have started to investigate whether or not linguistic properties are embedded in these models (Conneau et al., 2018; Liu et al., 2019; Tenney et al., 2019b; Hewitt & Manning, 2019). Then arguments began in this area. The frontline of these arguments is about *what probing models can we use to prevent them from learning the task itself*. While early practices and preliminary methods suggested on strictly linear probes (Alain & Bengio, 2017; Hewitt & Manning, 2019; Liu et al., 2019), Hewitt & Liang (2019) instead proposed control tasks that penalizes models being ability to learn the task itself, and had attempts on several Deep MLPs. Furthermore, Pimentel et al. (2020a) admitted this trade-off and took probing as an accuracy-complexity two-goal optimizing problem, and most radically, Pimentel et al. (2020b) insisted that probes should be as deep and complex as possible since they used them as estimators for $\mathcal{V}$-Information (Xu et al., 2020). Apart from disputes, there are also alternative theories proposed by

the researchers, like the code-description-length theory by Voita & Titov (2020) and the architectural bottleneck principle by Pimentel et al. (2022). These theories can be seen as patches under the supervised probing context since they're also addressing the complexity *vs.* accuracy tradeoff.

Apart from supervised probes, there are also parameter-free probes taking attention scores as evidence for dependencies (Clark et al., 2019; Vig & Belinkov, 2019; Ravishankar et al., 2021), resulting in individual dependency arcs or unlabeled dependency trees. If we take a broader view, we'll also find parameter-free activation explanation methods for more general-purpose concepts in deep learning research (Mu & Andreas, 2020; Antverg & Belinkov, 2022), together with some supervised methods (Radford et al., 2019; Lakretz et al., 2019; Dalvi et al., 2019). These methods, also called neuron analysis methods, were systematically evaluated by a recent work (Fan et al., 2023). We'll systematically compare our methods with these baselines in 4.

Similarly, there are also methods for extracting syntax rules from sentences without annotations, ranging from probability-based methods (Klein & Manning, 2002) to neural-network approaches (Shen et al., 2018). While achieving different tasks, they are essentially another side of a coin within the field of computational syntactics.

## 3 IPBP METHODOLOGY

To foster understanding, we'll first break our method into key points in the first section, and then introduce the details.

### 3.1 KEY ASPECT ANALYSIS

Given a sentence $X = x_1 x_2 \ldots x_n$ from dataset $\mathcal{D}$ and an arbitrary token pair $\langle x_i, x_j \rangle$ from that sentence, we define $l^{[i][j]}$ as the variable (also an element in the dependency tree adjacency matrix) for which kind of dependency exists from $x_i$ to $x_j$. $l^{[i][j]}$ can be a specific dependency type like nsubj, or $\phi$ if there's no dependency. If the sentence is fed into a transformer LM, there will be a series of attention scores matrices. An element in a specific matrix is denoted as $a_{b,h}^{[i][j]}$, which stands for the attention score from the $i$-th token to the $j$-th token in the attention matrix from the $h$-th attention head of the $b$-th transformer block. If we gather observations $l^{[i][j]}$ and $a_{b,h}^{[i][j]}$ for each token pair in the dataset, we'll get two co-occurring dataset-wide variables, $L$ and $A_{b,h}$, standing for the dependency and attention score of head $(b, h)$ at *any* token pair. Therefore, the goal of our probing can be divided into two:

- **MI Estimation**: For any head $b, h$, estimate the mutual information (MI) between $L$ and $A_{b,h}$ (denoted by $\mathrm{MI}(L; A_{b,h})$).

- **Tree Reconstruction**: A method for deriving a full dependency tree based on attention scores $A_{b,h}$.

Specifically, since $L$ is a discrete variable and $A_{b,h}$ is continuous, the joint distribution is a mixture distribution, the formula of MI is as follows (slightly different from classical definition):

$$\mathrm{MI}(L; A_{b,h}) = \sum_{l \in \mathcal{L} \cup \phi} \int f(l, a) \log \frac{f(l, a)}{P(l) f(a)} \mathrm{d}a \tag{1}$$

Where $\mathcal{L}$ stands for the set of all dependency relationships {nsubj, dobj, ...} and $f(l, a)$, $P(l)$, $f(a)$ is short for the density value of joint distribution $f(L, A_{b,h})$ at $L = l, A_{b,h} = a$, density of marginal distribution $f(A_{b,h})$ at $A_{b,h} = a$, and scalar probability $P(L = l)$.

Moreover, the second goal can be regarded as a Bayesian inference process taking $A_{b,h}$ as the *evidence* and $L$ as the *hypothesis*. The posterior distribution ($f(L = l | A_{b,h} = a)$) is required for tree reconstruction. Therefore, the key to achieving these two goals are those probabilistic distributions.

In the following sections we'll dive deep into how we can infer these distributions from the dataset.

### 3.2 GETTING THE DISTRIBUTIONS

**Initialization.** Assume the dataset $\mathcal{D}$ is already annotated with a series of *<sentence, dependency tree>* pairs, and a model with $\mathfrak{b}$ blocks and $\mathfrak{h}$ attention heads within each block. We first initialize a series of attention score sets $\mathcal{A}_{b,h;l}$ where $b \in \{1 \ldots \mathfrak{b}\}$, $h \in \{1 \ldots \mathfrak{h}\}$ and $l \in \mathcal{L} \cup \phi$. $\mathcal{A}_{b,h;l}$ means all possible attention scores of attention head $b, h$ between token pairs having dependency $l$.

**Gathering attention scores.** We iterate over the dataset and for a specific sentence $X \in \mathcal{D}$, we feed $X = x_1 \ldots x_n$ into the model, and for any token pair $\langle x_i, x_j \rangle$ $(i, j \in \{1 \ldots n\})$, the dataset provides its dependency relationship $l^{[i][j]}$ and the model provides the attention scores $a_{b,h}^{[i][j]}$ ($\forall b, h$). We add $a_{b,h}^{[i][j]}$ to the corresponding attention score set $\mathcal{A}_{b,h;l^{[i][j]}}$. After the iteration, all attention score sets will have all possible attention scores in the dataset.

**Getting the distributions.** After gathering attention scores, we'll estimate those required probabilities. The most intuitive one might be $P(L = l)$, since we can take the empirical probability $\hat{P}(L = l) = \frac{|\mathcal{A}_{b,h;l}|}{\sum_{l' \in \mathcal{L} \cup \{\phi\}} |\mathcal{A}_{b,h;l'}|}$, $\forall b, h$ (the number proportions) on the dataset as the approximate value. The tricky ones are the continuous probabilities. Given we already have abundant attention score samples, and the mixture distribution render requirements for only univariate densities, it's nice to use Kernel Density Estimation (KDE) to estimate them. Specifically, for every possible $\mathcal{A}_{b,h;l}$, we regard it as the observation of the attention variable $A_{b,h}$ under the circumstance of $L = l$. The samples in $\mathcal{A}_{b,h;l}$ follow the conditional density of $f(A_{b,h}|L = l)$. Using the Gaussian kernel and taking a specific bandwidth $B$ (See D.1), the kernel density $\hat{f}(A_{b,h}|L = l)$ can be estimated as:

$$\frac{1}{|\mathcal{A}_{b,h;l}| \cdot B} \sum_{i=1}^{|\mathcal{A}_{b,h;l}|} \frac{1}{\sqrt{2\pi \cdot \sigma_{\mathcal{A}_{b,h;l}}}} \exp\left(-\frac{x_0 - \mathcal{A}_{b,h;l}^{(i)}}{B}\right)^2 \tag{2}$$

where $\sigma_{\mathcal{A}_{b,h;l}}$ is the standard deviation of $\mathcal{A}_{b,h;l}$, and $\mathcal{A}_{b,h;l}^{(i)}$ means the i-th value of $\mathcal{A}_{b,h;l}$.

Again, if we take the view of Bayesian inference, with $A_{b,h}$ as evidence and $L$ as hypothesis, then the estimated $\hat{f}(A_{b,h}|L)$ is essentially the *likelihood density*. Applying the Bayesian theorem, we'll get the following equation:

$$f(L|A_{b,h}) = \frac{f(A_{b,h}|L)P(L)}{f(A_{b,h})} = \frac{f(A_{b,h}, L)}{f(A_{b,h})} \tag{3}$$

This means that given the likelihood $\hat{f}(A_{b,h}|L)$ and the prior $\hat{P}(L)$, we can multiply $\hat{f}(A_{b,h}|L)$ with $\hat{P}(L)$ to get the joint densities $\hat{f}(A_{b,h}, L)$. Moreover, by summing over all possible $L$s, we can estimate marginal density $\hat{f}(A_{b,h})$, and then the posterior probability $\hat{f}(L|A_{b,h})$ is computable. This resolves the computation of all the required probabilities as mentioned in Section 3.1.

### 3.3 ESTIMATING MI

With all these distributions, we're able to proceed to our two main goals: *MI estimation* and *Tree Reconstruction*. However, if we reexamine the MI formulation in Equation 1, we'll find that the $\text{MI}(L; A_{b,h})$ in Equation 1 measures how much shared information head $\langle b, h \rangle$ has about *every possible dependencies*. However, a more probable case might be that head $\langle b, h \rangle$ is only responsible for *certain dependencies*. This kind of *specialist* head is also the assumption of previous attention-analysis research like (Htut et al., 2019). Even though such versatile heads exist, an MI corresponding to all dependencies is still too coarse-grained. Therefore, it is necessary to tweak Equation 1 to make the MI formulation fit this *specialist* assumption. The new formulation is as follows:

$$\text{MI}_{\text{binary}}(l; A_{b,h}) = \int f(l, a) \log \frac{f(l, a)}{P(l)f(a)} \mathrm{d}a + \int f(\neg l, a) \log \frac{f(\neg l, a)}{P(\neg l)f(a)} \mathrm{d}a \tag{4}$$

In that equation, $f(\neg l, a)$ is short for the density value of $f(\neg l, A_{b,h})$ at $A_{b,h} = a$, where $f(\neg l, A_{b,h})$ stands for the joint density between all dependencies other than $l$ and $A_{b,h}$. In practice, it can be

gained by marginalizing $\hat{f}(A_{b,h}, L)$ over all possible $L \in (\mathcal{L} \cup \{\phi\}) - \{l\}$. $P(\neg l)$ stands for the possibility of dependencies other than $l$, which can be estimated using $1 - \hat{P}(l)$.

### 3.4 GETTING HIGHLY SYNTACTICAL HEADS

By now, having posterior distributions $\hat{f}(L|A_{b,h})$ and $\text{MI}_{\text{binary}}$ feasible for estimating the independent importance of each dependency type, it's possible to reconstruct the trees. The basic idea of our tree reconstruction is: for every dependency relationship $l$, we pick out attention heads highly responsible for $l$, constituting the head set $\mathcal{H}_l$. We then infer the possibilities of dependency arcs of $l$ jointly decided by the posteriors of heads from $\mathcal{H}_l$ by using $\text{MI}_{\text{binary}}$ to balance between posteriors of each head from $\mathcal{H}_l$, forming the overall possibility for a dependency arc with relation $l$. Finally, we use a decoding algorithm to build the dependency tree based on these overall possibilities.

To form $\mathcal{H}_l$, it's natural to think about setting a threshold on $\text{MI}_{\text{binary}}(l; A_{b,h})$. However different dependency relationships might have different $\text{MI}_{\text{binary}}$ magnitudes, so an adaptive threshold conditioning on specific relations is necessary. Remind the fact that mutual information is upper-bounded by the individual entropies of each random variable (in our case, $H(\mathbf{1}_l(L))$ and $H(A_{b,h})$, where $\mathbf{1}_l(L)$ denotes $L$ equals to $l$ or not). Since $A_{b,h}$ is continuous, and the entropy analogs of continuous variables (variational entropies) are known as incorrect analogs (and sometimes negative), this makes it unable to act as an upper bound. Therefore, we choose to estimate $H(\mathbf{1}_l(L))$ as:

$$\hat{H}(\mathbf{1}_l(L)) = \hat{P}(L) \log \hat{P}(L) + \hat{P}(\neg L) \log \hat{P}(\neg L) \tag{5}$$

If MI is divided by the entropy, the resulting proportions ($\frac{\text{MI}_{\text{binary}}(l; A_{b,h})}{\hat{H}(\mathbf{1}_l(L))}$) will be normalized into $[0, 1]$, which can act as the adaptive threshold.

### 3.5 TREE-RECONSTRUCTION ALGORITHM

After getting $\mathcal{H}_l$s, another problem occurs: As mentioned before, previous probing practices mainly aim at building unlabeled trees. Even those supervised dependency parsing methods (Dozat & Manning, 2017; Tian et al., 2022) rely on separate networks for predicting arcs and labels. Therefore, these methods are operating on a simple probability space with only probabilities on *the existence of dependency arcs*, and then individual probability spaces for the label of each arc. Moreover, their methods only involve one or two networks responsible for predicting probabilities, while our method instead has bunches of posterior probabilities. Therefore, it's necessary that we design a decoding algorithm that not only *balances each posterior* but also constitutes a *valid probability space*.

We first make an assumption that the overall possibility of dependency arcs can be individually conditioned on each head in $\mathcal{H}_l$. Theoretically, a good model to balance each posterior is to treat the prediction of dependency arcs as a voting process: for dependency $l$, each head $\langle b_i, h_i \rangle \in \mathcal{H}_l$ can be seen as a participant with a representation of $e^{\text{MI}_{\text{binary}}(l; A_{b,h})}$. The probability of a dependency arc of $l$ can be seen as the probability of a total representation of approval larger than a proportion (like half or two-thirds) . However, due to the non-discrete weights, the problem cannot be efficiently dynamically programmed, resulting in a search space of $\mathcal{O}(2^{|\mathcal{H}_l|})$, which will be terribly inefficient during inference. Instead, we relax this voting problem to an easy-computing yet rational form: We take the geometric mean of the posteriors. Specifically, let $\text{GP}_{\mathcal{H}_l}(x_i, x_j; l)$ be the geometrically-averaged probability of an arc of $l$ between tokens $x_i$ and $x_j$ conditioned on heads in $\mathcal{H}_l$. In logarithmic space, the geometric mean is:

$$\log \text{GP}_{\mathcal{H}_l}(x_i, x_j; l) = \frac{\sum_{\langle b_k, h_k \rangle} \text{MI}_{\text{binary}}(l; A_{b_k, h_k}) \cdot \log \hat{f}(L = l | A_{b_k, h_k})}{\sum_{\langle b_m, h_m \rangle} \text{MI}_{\text{binary}}(l; A_{b_m, h_m})} \tag{6}$$

This is approximately equivalent to the Logarithmic Opinion Pooling (Heskes, 1997) technique widely adopted in Bayesian inference, thus acting as a reasonable approximation when the number of experts (in our case, heads in $\mathcal{H}_l$) is relatively large. Given this good approximation, it still has a problem: if we sum over all probabilities of each possible dependency relationship (in our $\text{MI}_{\text{binary}}$ case, $l$ and $\neg l$), it is not guaranteed to be 1, and different kind of dependency arcs are decided by different sets of

heads, recalling the problem of a valid and homogeneous probability space. To resolve this, we build a larger multivariate probability space of $\{0,1\}^{|\mathcal{L}|+1}$. We take the voting process of the dependency between $x_i$ and $x_j$ as $|\mathcal{L}| + 1$ independent votes. The $l$-th ballot votes for the existence of the $l$-th dependency from $\mathcal{L}$, using the $\mathrm{GP}_{\mathcal{H}_l}(x_i, x_j; l)$ in Equation 6 as the probability of *existence*, and $1 - \mathrm{GP}_{\mathcal{H}_l}(x_i, x_j; l)$ as the probability of *non-existence*. The overall probability $P(x_i, x_j; l)$, meaning the probability of an arc of $l$ between tokens $x_i$ and $x_j$ conditioned on *all* highly responsible heads $\mathcal{H}_1 \cup \cdots \cup \mathcal{H}_{|\mathcal{L}|} \cup H_\phi$, is calculated as follows:

$$P(x_i, x_j; l) = \mathrm{GP}_{\mathcal{H}_l}(x_i, x_j; l) \ \times \prod_{l' \in \mathcal{L} + \{\phi\} - \{l\}} \{1 - \mathrm{GP}_{\mathcal{H}_{l'}}(x_i, x_j; l)\} \tag{7}$$

This results in the decision of all kinds of dependency arcs in a unified and valid probability space. By now, the two problems introduced by *multi-head* and *multi-label* are solved. We're only need a decoding algorithm utilizing the overall probabilities to arrive at complete trees. Specifically, following previous supervised dependency parsing methods, we're using the Eisner dynamic programming algorithm (Eisner, 1996) as the decoding algorithm. Readers might refer to Appendix D.1 for implementation details of our methods, like hyperparameters and our GPU-optimized KDE and integral methods.

### 3.6 Is IPBP trivial?

Since MI estimation is a small hot topic in statistics, it's helpful to perform methodological comparisons with related methods. We found two methods sharing (minor) principles with our method: The first one (Moon et al., 1995) is a method for estimating MI between two observations within a time series using KDE. They did three separate KDEs, with one multivariate one. While it's a known issue that KDE quickly becomes inferior when there is more than one variable, known as the *dimensionality curse*, their method is inevitably introducing errors (and also unapplicable to our attention-dependency mixure distribution setting). We instead dexterously circumvented the dimensionality curse, also making the least number of estimations possible (limited to 1) by exploiting mixed-joint distribution and Bayesian theorems.

Another one also focuses on mixed joint distribution (Gao et al., 2017). However, they use a $k$NN-like algorithm to estimate point-wise mutual information (PMI) and average it over the dataset. Their method didn't provide any valid probability distributions, thus offering no possibility of tree reconstruction, and also allowing less chance for dataset-level or visualization-based explanations.

## 4 Experiments

In this section, we're going to systematically compare our method with a series of probing as well as neuron analysis baselines.

### 4.1 Evaluation Strategy

We divide the baseline employed into two sets based on the two sub-tasks introduced in Section 3.1. For the first subtask **MI Estimation**, we introduce a series of head-selection baselines, replace the estimated MI with the importance introduced by these methods, keeping the tree-construction algorithm fixed, and compare the qualities of reconstructed trees. This essentially uses constructed trees as an external "proxy" for the quality of head-selection. Apart from that, we also employed the voting theory-based intrinsic comparison method introduced by (Fan et al., 2023). The core idea is that they compare neuron importance methods by *letting other methods vote on one method*. A neuron selection method with higher consistency with the rest is considered a preferred one. Since the high-importance head sets like $\mathcal{H}_l$ are of variable sizes, we instead compare the importance rankings of all heads given by two head-selection methods.

For the second subtask **Tree Reconstruction**, we compared our MI and posterior-based algorithm with the common practice of previous attention-based methods. We believe dividing into two groups of baselines is better for illustrating the individual contributions of each submodule of IPBP.

## 4.2 BASELINE METHODS

For head-selection baselines, we'll start from several strong neuron analysis methods evaluated by a recent paper (Fan et al., 2023):

**Probeless** (Antverg & Belinkov, 2022): This is a parameter-free method, which gets the correlation scores by calculating mean activation values corresponding to having/not having a concept alongside the dataset. In our situation, we use the following instead of $\text{MI}_{\text{binary}}$:

$$\text{PL}(l; A_{b,h}) = \sum_{l' \in \mathcal{L}+\{\phi\}-\{l\}} \left| \bar{\mathcal{A}}_{b,h;l} - \bar{\mathcal{A}}_{b,h;l'} \right| \tag{8}$$

Where $\bar{\mathcal{A}}_{\cdot,\cdot;\cdot}$ denotes the mean value of a specific attention score set. Note that despite its simplicity, this method is evaluated as the method being most consistent (thus most preferred) with others by Fan et al. (2023).

**IoU** (Mu & Andreas, 2020): This method uses Jaccard Similarity as a correlation criterion. In our implementation, we use the following form:

$$\text{IoU}(l; A_{b,h}) = \frac{|\mathcal{A}_{b,h;l} \cap [\tau, +\infty)|}{|\mathcal{A}_{b,h;l}| + \sum_{l' \in \mathcal{L}+\phi-\{l\}} |\mathcal{A}_{b,h;l'} \cap [\tau, +\infty)|} \tag{9}$$

Where $\tau$ is a threshold serving as selecting a salient score. Following the original authors, we set it to the top 99.5% value among values in $\mathcal{A}_{b,h;\cdot}$.

**The Linear Feedforward Family**: This method refers to a series of methods performing correlation ranking by training a supervised linear network $W_\theta$. Specifically, the equation below gives a uniform formulation of these methods:

$$W_\theta = \operatorname*{arg\,min}_{W_\theta \in \Theta} \sum_{X \in \mathcal{D}} \sum_{x_i, x_j \in X \times X} \log P_\theta(l = l^{[i][j]} | a^{[i][j]}_{1,1\ldots\mathfrak{b},\mathfrak{h}}) \tag{10}$$

Where $W_\theta$ is a matrix of shape $\mathfrak{b}\mathfrak{h} \times (|\mathcal{L}| + 1)$, and $a^{[i][j]}_{1,1\ldots\mathfrak{b},\mathfrak{h}}$ denotes the concatenated vector of attention scores between $x_i$ and $x_j$ for all attention heads, and $P_\theta(l^{[i][j]} | a^{[i][j]}_{1,1\ldots\mathfrak{b},\mathfrak{h}})$ stands for the probability of the ground-truth label estimated by the network. When $\lambda_1 = 1$, $\lambda_2 = 0$, this equation becomes Lasso (Radford et al., 2019), when $\lambda_1 = 0$, $\lambda_2 = 1$, it becomes Ridge (Lakretz et al., 2019), and $\lambda_1 = 1$, $\lambda_2 = 1$ corresponds to ElasticNet (Dalvi et al., 2019). We use ElasticNet as a representative. After gaining the trained $W_\theta$, we use the weight entry mapping attention score of head $\langle b, h \rangle$ to the probability of relation $l$ as the correlation value, $\text{LFF}(l; b, h)$.

$\mathcal{V}$-**Information**: Xu et al. (2020) proposed to use a trainable network as an approximation of conditional probabilities, and use the mean logarithm probabilities as approximations of conditional entropies based on the law of large number. This is the state-of-the-art mathematical-rigorous entropy estimation algorithm, used by previous methods also taking information-theoretic perspectives (Pimentel et al., 2020b; 2022). Specifically, in our case, we use $\max H_\mathcal{V}(l|A_{\cdot,\cdot}) - H_\mathcal{V}(l|A_{b,h})$ as replacement of $\text{MI}_{\text{binary}}(l; A_{b,h})$, and as the equation shows:

$$H_\mathcal{V}(l|A_{b,h}) = \frac{1}{\sum_{X \in \mathcal{D}} |X|^2} \sum_{X \in \mathcal{D}} \sum_{\langle x_i, x_j \rangle \in X} \tag{11}$$

$$\mathbf{1}_{\{l\}}(L^{[i][j]}) \log \text{MLP}_{b,h;l}(a^{[i][j]}_{b,h}) + \mathbf{1}_{\mathcal{L}+\{\phi\}-\{l\}}(L^{[i][j]}) \log \left[ 1 - \text{MLP}_{b,h;l}(a^{[i][j]}_{b,h}) \right]$$

Where $\text{MLP}_{b,h;l}(\cdot)$ are deep MLPs individually trained using head $\langle b, h \rangle$ to predict label $l$.

Under each head-selection setting, for fair comparison, we set a limit of the total number of syntactical heads $\sum_{l \in \mathcal{L} \cup \{\phi\}} |\mathcal{H}_l|$ of 2000 for every head-selection method.

For the tree-construction alternative, we use **raw attention score**: Under this setting, we still use MI as the head importance criteria, but for a specific head $\langle b, h \rangle \in \mathcal{H}_l$, we use the attention score

$a_{b,h}^{[\cdot][\cdot]}$ instead of the posterior $\hat{f}(L|A_{b,h})$ in the reconstruction algorithm, with all other formulations (balancing heads and forming valid probability space) unchanged. This simple intuition is the common underlying principle of previous methods attention probing methods (Clark et al., 2019; Vig & Belinkov, 2019; Ravishankar et al., 2021). We found that due to the absence of our estimated posteriors, if $\sum_{l \in \mathcal{L} \cup \{\phi\}} |\mathcal{H}_l|$ reaches 2000, the scores of all heads will be rather noisy. Therefore, we choose to select top-$k$ heads based on MI for each label. We did a grid search and found the top-8 setting has ideal performance.

Moreover, we also introduced two simple baselines, **left/right branching** assuming the dependency head of each word is always the previous/latter word (this results in unlabeled dependency trees), and **random model** that applies IPBP on a model with the same structure but with its weights randomly initialized. These methods are mainly used to rule out the worst situations where our methods (also the baselines) fall behind the simplest baselines.

### 4.3 MODEL, DATASET AND METRICS

We applied our methods on `open_llama_7b`(Geng & Liu, 2023), `Meta-Llama-3-8B`(AI@Meta, 2024), `vicuna-7b`(Chiang et al., 2023), and `Mistral-7B`(Jiang et al., 2023). These models are decoder-based LLMs consisting of a great many layers and heads. Compared with pre-trained models like BERT (Devlin et al., 2019) or XLNet which are widely used in previous probing research, these LLMs' attention heads might have rather varied functionalities, offering more insights under the contemporary LLM research context.

With respect to collecting attention scores, we cache the Q/K/V of each attention head and use them to re-calculate the unmasked attention scores. This is mainly because, as sentence lengths vary, the softmax-normalized attention scores also vary in magnitude, resulting in attention scores from sentences of different lengths falling in different intervals, which undermines the density estimation process. (This is especially true for masked attention matrices since they have rows of varying numbers of unmasked elements, so even attention scores within the same sentence vary in magnitudes.) Given that softmax is not bijective, using cached QK to reconstruct the unnormalized scores is inevitable. Since directly using the reconstructed attention matrix might involve using masked attention scores for estimation, we talked about two structural alternatives of our method in Section 4.4 to discuss potential effects. To deal with subword tokenization, we use the attention scores on the last sub-token of each word.

For datasets, we use the English, French, and Spanish partitions of Universal Dependencies (UD) 2.9 (Zeman et al., 2021) as our dataset. Dataset statistics are shown in Appendix D.1.

For evaluation metrics on reconstructed trees to evaluate the quality of constructed trees, like supervised dependency parsing methods, we also employ labeled attachment scores (LAS), and unlabeled attachment scores (UAS) (different from previous probing methods, which only used undirected unlabled attachment scores (UUAS)). For intrinsic comparison, since each head-selection method gives a $[|\mathcal{L}|, \mathfrak{b} \times \mathfrak{h}]$ matrix, for any two head-selection methods, we calculate the Spearman-R with respect to the corresponding rows for each dependency label, and average them across labels.

Note that, unlike IPBP, some baselines are rather computationally prohibitive. Given the observations that most models have similar IPBP tree-extraction performance (Section 4.5) and similar performances when applied on our multilingual dataset (Appendix B), also considering the vast number of model-language-baseline compositions, experiments of baseline comparison are mainly conducted on the English dataset using model `open_llama_7b` (it seemed most "familiar" with our dataset as shown in Appendix B, acting as a good representative)

### 4.4 IPBP STRUCTURAL ALTERNATIVES

Apart from comparing with previous methods, we're also curious about our model's designs. Therefore, we propose several alternative structures:

**Positive MI**: We noticed that the attention score samples exhibit a long-tail characteristic: most samples come from $\mathcal{A}_\phi$, since most pairs of words don't have dependency arc in between. $\mathcal{A}_\phi$ might be noisy, consisting of various non-syntactic inter-token relationships, and MI estimations based on samples in $\mathcal{A}_\phi$ might be affected by this long tail noisy distribution. Other score sets $\mathcal{A}_1, ... \mathcal{A}_{|\mathcal{L}|}$

are having approximately the same magnitudes and their corresponding token pairs are guaranteed to have any dependency relationship. Therefore, we also calculated a more syntactical MI, namely $\text{MI}_{\text{pos}}$, with the following formulation:

$$\text{MI}_{\text{pos}}(L; A_{b,h}) = \sum_{l \in \mathcal{L}} \int f_{\text{pos}}(l, a) \log \frac{f_{\text{pos}}(l, a)}{P_{\text{pos}}(l) f_{\text{pos}}(a)} \tag{12}$$

In that equation, $P_{\text{pos}}(\cdot)$, $f_{\text{pos}}(\cdot, \cdot)$ actually stand for conditional possibilities when $l \neq \phi$, estimated by $\hat{P}_{\text{pos}}(L = l) = \frac{|\mathcal{A}_{b,h;l}|}{\sum_{l' \in \mathcal{L}} |\mathcal{A}_{b,h;l'}|}$ and $\hat{f}_{\text{pos}}(L, A_{b,h}) = \hat{f}(A_{b,h}|L)\hat{P}_{\text{pos}}(L)$. During implementation, we'll use a balance factor $\alpha$ and calculate the mixed MI $\text{MI}_{\text{mix}}(\cdot; \cdot) = \alpha \text{MI}_{\text{binary}}(\cdot; \cdot) + (1 - \alpha)\text{MI}_{\text{pos}}(\cdot; \cdot)$

**Arc First**: Unlike previous methods, we're directly obtaining labeled dependency trees, bypassing the process of dependency arc predicting. We're curious about whether it's a good choice. Under this setting, instead of estimating $\hat{f}(A_{b,h}|L)$, we'll directly estimate the unlabeled likelihoods $\hat{f}(A_{b,h}|L \in \mathcal{L})$ and $\hat{f}(A_{b,h}|L = \phi)$, and calculate the corresponding multivariate probabilities together with corresponding MI values. We'll compare UAS to check the quality of reconstructed unlabeled trees.

**Transposed**: Sometimes, we're unsure whether the attended token acts as a dependency head, or a dependant. Another thing to consider is that lower-triangular entries of the (reconstructed) attention matrix might correspond to dependencies that ought to correspond to the higher-triangular "useless" (masked) attention scores. So we let $l^{[i][j]}$ correspond to $a_{b,h}^{[j][i]}$, and repeat the whole IPBP process under this setting.

**Undirected**: This is a more radical leap towards eliminating attention score uselessness. Under this alternative, we relax the problem to probing undirected (but labeled) dependency trees through "flipping" the up-triangular area of the dependency adjacency matrix to the lower-triangular part (*i.e.*, $l_{[i][j]}$ corresponds to $a_{b,h}^{[i][j]}$ if $i > j$ else $a_{b,h}^{[j][i]}$), and perform the whole IPBP process. This results in being unable to decide dependency directions, as a tradeoff for using attention scores that are actually in-use during generation. Under this setting, undirected trees are reconstructed using the maximum spanning tree algorithm.

What's more, we've also done ablations regarding KDE bandwidth and posterior pooling methods. See Appendix E for details.

## 4.5 RESULT AND ANALYSIS

Results are shown in Table 1, 2, and 3. We can see that our method applies to multiple models and languages. We also observe that it gives better sets of selected heads (using trees as proxies and also intrinsically evaluated), and produces more faithful trees given the same set of heads, compared with all competitive baselines. To be noticed, we can find that supervised methods like LFF or $\mathcal{V}$-Information may still fall behind our statistical method. Considering methods like $\mathcal{V}$-Information require much larger computational budgets, and we have to use several tricks to adapt $\mathcal{V}$-Information to our attention probing settings (see Appendix D.2 for details), we argue that supervised deep networks might not be the panecea especially when the data is long-tailed or low-dimensional. For tree reconstruction alternatives, the well-used tree-reconstruction baseline raw-score, despite being specially treated, still has a great performance gap compared with our posterior-based method, further justifying its necessity. For structure alternatives, we notice that incorporating $\text{MI}_{\text{pos}}$ will give performance benefits, inspiring further improvements. The transposed setting will still capture a relatively smaller portion of dependencies, with undirected setting capturing better trees (ignoring slightly different metrics for simplicity) than it. For more intriguing conclusions regarding how models capture dependencies of different directions, please see Section 4.6. For those simple and random baselines, we find that they fall far behind most of our methods and baselines, indicating that IPBP and the baselines are producing non-trivial results. Finally, by comparing with our arc-based baseline, we'll find that we're actually at a triangular balance: we probed for more *accurate*, *labeled* trees using a more *straightforward* method: no need for individual arc probing.

Table 1: Results across models and languages

| Model/Language | UAS | LAS |
|---|---|---|
| Meta-Llama-3-8B | 39.9 | 26.6 |
| Mistral-7B | 50.4 | 30.9 |
| vicuna-7b | 41.5 | 25.8 |
| open_llama_7b | 49.1 | 30.6 |
| open_llama_7b+fr | 44.7 | 25.6 |
| open_llama_7b+es | 44.8 | 21.9 |

Table 2: Tree reconstruction results of our IPBP and different baselines. For undirected settings, results* are UUAS and ULAS instead.

| Method | UAS | LAS |
|---|---|---|
| Random Model | 12.4 | 0.8 |
| L. Branching | 16.6 | - |
| R. Branching | 26.4 | - |
| Probeless | 34.8 | 20.9 |
| IoU | 38.3 | 26.6 |
| ElasticNet | 41.9 | 31.3 |
| $\mathcal{V}$-Information | 41.3 | 20.9 |
| Raw Score | 32.3 | 16.6 |
| IPBP | 49.1 | 30.6 |
| IPBP (transposed) | 42.6 | 28.0 |
| IPBP (undirected) | 45.3* | 28.4* |
| IPBP + MI$_{pos}$ | **49.9** | **34.8** |
| IPBP (arc only) | 36.5 | N/A |

## 4.6 FURTHER ANALYSIS

Like previous probing methods, we'll also do fine-grained analysis of our reconstructed trees and estimated MI values. Instead of listing up MI values and doing trivial analyses, we decided to provide several intriguing and informative conclusions, giving inspiration to those upcoming research.

The first conclusion is that **decoder models adaptively capture left/right dependencies**: Since the masked decoder attention can only attend to previous tokens, it's reasonable that left dependencies (dependencies pointing to front words) can be well captured. What makes it more intriguing is that right dependencies (pointing to rear words) might also be captured in a "attend-back" manner. We draw this conclusion by comparing the top-10 most well-reconstructed labels between original IPBP and the transposed alternative. We find that there's more look-ahead dependencies (5 of 10) under the transposed setting compared with the original setting (3 of 10). The second conclusion is that **model layers correspond to tree layers** to some extent: lower layers are for local/phrasal dependencies, while higher layers are for global/sentence-wide dependencies. This corresponds to intuitions but was never systematically justified before due to the lack of MI-like criteria and overfocusing on unlabeled trees. Thanks to the fine-grained MI, similar to the "Center-of-Gravity" introduced by Tenney et al. (2019a), we can calculate the MI-weighted layer indices $\frac{\sum_{h \in \{1...\mathfrak{h}\}} \sum_{b \in \{1...\mathfrak{b}\}} \mathrm{MI}_{pos}(A_{b,h};l) \cdot l}{\sum_{h \in \{1...\mathfrak{h}\}} \sum_{b \in \{1...\mathfrak{b}\}} \mathrm{MI}_{pos}(A_{b,h};l)}$ for each label, where smaller weighted indices indicate dependencies having more lower-layer heads responsible for it. Among these top-10 labels, we calculated the Pearson correlation coefficient $\rho$ between the weighted layer index and average depth (maximum distance to leaf nodes) of each dependency label, getting a result of 0.69 with $p = 0.03$ for a null hypothesis of no correlation. Apart from these, we also drew some other informative conclusions by a series of further MI/distribution analysis methods, including visualization, mechanistic analysis, and unbiased (not specifying "things to probe is dependency" in advance). These results further ensure the faithfulness of IPBP and give intriguing conclusions and insights. Please refer to Appendix A for details.

## 5 CONCLUSION

We proposed a method that can estimate MI and reconstruct labeled dependency trees without introducing any trainable networks. Indeed, our method is achieving an "impossible triangle": it has *simpler* architectures requiring small computation budgets, while producing more *complex* and *high-quality* trees, and also *transparent for explanation*, meaning that researchers can get fine-grained head-level MI estimation, and a bunch of intuitive probability functions, without worrying about *did my network furtively learn the task*? Through comparing with a series of competitive baselines, we ensured its effectiveness, and then made several informative conclusions based on our estimated MI and reconstructed trees. The number of conclusions is limited due to the content limit. Since our method provides an analytical backbone, we strongly encourage future fine-grained research based on the fine-grained MI values and distributions of our method.

ACKNOWLEDGMENTS

This research is supported by National Key R&D Program of China (No. 2024YFF0907003) and Beijing Natural Science Foundation (L247010). We'd also thank the anonymous reviewers for their efforts and valuable suggestions.

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

# A   VARIFYING THE FAITHFULNESS OF IPBP

While IPBP produces probability distributions and MI between a great many attention head-dependency label compositions, it is hardly ever possible (also inefficient) to list or visualize all MIs/distributions. So far, we mainly discussed tree extraction results, which naturally raises the doubt that *is the high MI expert heads just "happened" to have high MI*? We'll investigate this question by showcasing other fine-grained (also page-consuming) conclusions that may otherwise be exhibited after Section 4.6.

## A.1   VISUALIZATION OF ESTIMATED PROBABILITY DISTRIBUTIONS

The most thrilling one we find might be the apparent **dual-peak** pattern among the $\phi$-joint densities of heads in $\mathcal{H}_l$: Figure 1 displays the joint distributions ($\hat{f}(A_{b,h}, l)$) of the attention heads whose $MI_{\text{binary}}$ rank top-5 and last-5, for $L \in \{\text{pobj,nsubj}\}$ (these are two well-captured dependencies). From that figure we might notice an apparent distributional pattern of these expert heads: the $\phi$-joint densities $\hat{f}(\neg l, A_{b,h})$ of these heads tend to have an apparent *sub-peak*, and these sub-peaks *overlap* with the main peaks of these $\hat{f}(l, A_{b,h})$ distributions, while those of non-expert heads don't (expect for head 29,21 and 15,29, which are expert heads for other dependencies by coincidence). While the non-expert heads exhibit approximately Gaussian-like densities, one seemingly plausible explanation of this "unexpected sub-peak" might be that, the high-MI expert heads do have the bahaviour of attending to words having that dependency, while the sub-peaks correspond to the cases where the head *thought* the token pair was of that dependency, but doesn't. The "errors" that expert heads made instead become the evidence for their behaviors.

## A.2   MECHANISM-BASED CASE STUDIES

Another conclusion is derived via a mechanistic interpretability perspective. We ask the following questions: If an expert attention head filtered by MI is expected to be highly responsible for a certain dependency, in real forward passes, will that attention head give human-understandable and interesting effects to model behavior?

Table 3: Intrinsic evaluation (label-averaged Spearman-R) between each pair of head-selection methods. Higher value means higher consistency with other methods. Values are averaged for each method in the last row.

|  | Probeless | IoU | Elasticnet | IPBP |
|---|---|---|---|---|
| Probeless | 1 | 0.198 | 0.276 | 0.327 |
| IoU | 0.198 | 1 | 0.224 | 0.438 |
| ElasticNet | 0.276 | 0.224 | 1 | 0.427 |
| IPBP | 0.327 | 0.438 | 0.427 | 1 |
| Column Average (w.o. diagonal) | 0.267 | 0.286 | 0.309 | 0.398 |

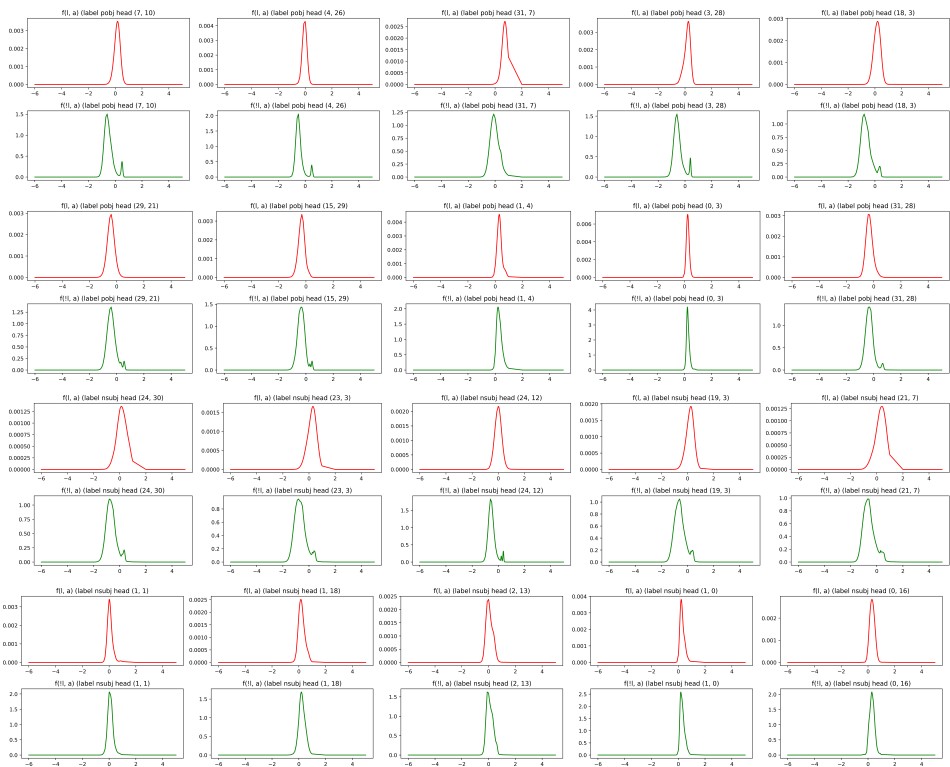

Figure 1: Estimated joint densities $(f(l, A_{b,h}), f(\neg l, A_{b,h}))$ of heads with top-5/last-5 MI, when $l \in \{\text{nsubj, pobj}\}$. The upper half is for dependency pobj, while the bottom half is for dependency nsubj. For each dependency, the first ten plots are joint densities of attention heads with top-5 MI, with the first five plots for $\hat{f}(l, A_{b,h})$ and last five plots for $\hat{f}(\neg l, A_{b,h})$. Correspondingly, the last ten plots are joint densities of attention heads with last-5 MI.

To investigate this, we proposed a "head ablation" technique: for a specific expert attention head, during the forward pass, we set the value vectors of MHSA to zero. Since attention heads operate on the residual stream by taking the weighted sum of value vectors, zeroing out the value states is equal to totally removing one attention head. This is also equal to cutting off the "OV-circuit" (proposed by (Elhage et al., 2021)). It also share some principles with the "activation patching" technique of many mechanism interpretability practices (Wang et al., 2022). The most observable model behavior (also well-used by previous mechanistic research) is the next-token logit, so we compare the logits after ablating the top-5 expert heads of a specific dependency with the original logits. Moreover, in order to exclude possible effects of the ablation itself, we also performed a series of random ablations as the control group (we ablated several times, each time five random heads were ablated, and we took their average logits).

The noticeable result is that, for a specific dependency label $l$, its expert heads particularly add to the logits of *words that require the model to understand $l$*. One typical example is shown as follows: for the incomplete sentence *"In order to protect the environment, eco-friendly industries were"*, for the tokens having top-10 logits among the predicted next tokens, the original logits / logits after ablating amod's top-5 transpose (amod is a right dependency) expert heads, and the average random ablation logits are shown in Table 4:

Table 4: (Unnormalized) next-token logits for sentence *In order to protect the environment, eco-friendly industries were* before/after top-5 head ablation and random head ablation. ↓ means the logits has decreased, and vice versa.

| Next Token | Original Logits | Logits w/o Top-5 Expert Heads | Avg. Logits After Random Ablation |
|---|---|---|---|
| established | 5.75 | 5.69↓ | 5.88↑ |
| developed | 5.72 | 5.69↓ | 5.75↑ |
| introduced | 5.00 | 4.94↓ | 4.97↓ |
| encouraged | 4.88 | 4.72↓ | 5.06↑ |
| set | 4.81 | 4.72↓ | 4.97↑ |
| started | 4.47 | 4.47 | 4.44↓ |
| promoted | 4.38 | 4.28↓ | 4.47↑ |
| required | 4.13 | 4.03↓ | 4.22↑ |
| born | 4.03 | 4.13↓ | 3.97↓ |
| formed | 4.00 | 4.00 | 4.03↑ |

We might notice an interesting phenomenon: these top-10 words basically express positive actions on eco-friendly industries. After ablating several `amod` expert heads, most of the logits decreased, while logits didn't exhibit unified increase/decrease behavior after random ablations. The difference shows that the decrease of logits is dedicated to ablating `amod` expert heads. This can be explained intuitively: if there weren't the adjective modifier "eco-friendly", considering the context "to protect the environment", common-purpose "industries" (probably polluting) should be restricted or contained. Only the "eco-friendly" industries should be "established" or "promoted". Therefore, the appearance of such words is highly dependent on understanding that "eco-friendly" modifies "industries", *i.e.*, the `amod` dependency.

## A.3 UNBIASED DEPENDENCY SYNTAX DISCOVERY

Sometimes people may argue, "probing assumes a syntactic structure (like the dependency syntax trees) in advance, it is doubtful whether specific syntactical relationships really aside in model's internal mechanism, or the researchers are fabricating a *Texas sharpshooter fallacy*.

We think a good solution to this doubt is to design an *unbiased* strategy: a strategy that does not assume any concepts in advance and make its assumptions only based on the model structure itself. We take the unbiased assumption of "maximum attention weight": for each token position, the token that an attention head gives most attention weight to can be considered as a token forming a relationship that the attention head favors. We designed the "maximum attention label recall", that is, for each dependency relationship, we calculate the count that an attention head assign the maximum attention weight to the dependency head/dependant (depending on which token comes earlier, since we adopt the first conclusion in Section 4.6), divided by the total number of dependency token pairs of that relation over the dataset. Note that we use the term and concept "recall" since dependencies are sparse over the dataset and on tokens not having such dependency the attention head might achive something else or just attend to a "no-op" position just like the first token. For each dependency relationship, we'll find an attention head having most maximum attention label recall. We then analyse how well the results from this unbiased strategy align with the results of IPBP by calculating the MI rank of head having most recall among all heads for each dependency. Results are shown as follows in Table 5.

Table 5: Maximum attention-based unbiased attention head analysis results for dependency relationships having top-10 largest maximum attention label recalls.

| Recall Rank | Label Name | Maximum Attn. Label Recall | MI Rank |
|:---:|:---:|:---:|:---:|
| #1 | root | 100.00 | 3 |
| #2 | csubjpass | 100.00 | 329 |
| #3 | iobj | 95.24 | 5 |
| #4 | expl | 93.94 | 3 |
| #5 | possessive | 92.63 | 35 |
| #6 | prt | 92.24 | 53 |
| #7 | mwe | 91.07 | 10 |
| #8 | auxpass | 86.12 | 1 |
| #9 | pobj | 85.55 | 1 |
| #10 | det | 82.70 | 1 |
| N/A | Average of all Labels | 57.25 | 66 |

We can draw two exciting conclusions from the results. First, **attention heads do capture dependency syntax even under simple unbiased assumptions**. We can find that the top-10 recalls are rather high, with the average maximum recall still more than half. Many of the top-10-recall dependencies are frequent and common dependencies, meaning that there are for most of the attention pairs in real-world data, there'll be heads that covering it by assigning a maximum attention score. Second, **maximum attention score aligns well with IPBP MI**. For the top-10-recall dependencies, the attention head that contribute the maximum label recall are mostly heads that are highly-ranked by MI (except for nsubjpass which is rare in the dataset). The average rank is also among the top 10% (open\_llama\_7b has 1024 attention heads in total). The unbiased probing results and comparisons towards our methods reveal the Texas sharpshooter concern to a large extent.

## B  LLM ABILITIES COMPARISONS AND ANALYSIS

Since our experiments are conducted on multiple models and languages, it is necessary to investigate the (possibly multilingual) performance differences of the models we've tested.

### B.1  METHODS

A common intuition is that there are open benchmarks [1] for LLMs' generic abilities and each LLM's research paper also reported relevant performances. However, the UD dataset is syntactically (also lexically) complex compared with most common-purpose LLM benchmarks. Therefore, it is required that we test the LLMs' abilities on our dataset.

While treebanks are sentences, a direct way of testing LLM capabilities on such datasets is to use perplexity (PPL). However, we indicate that PPL can be used to compare language model abilities for the same model, but might not yield fair results when comparing *across* models: a simple example is that for a sentence $X = x_1 x_2 ... x_n$, for each position $i \in [1...n]$, if two models $p_1(\cdot | \cdots ; \theta_1)$ and $p_2(\cdot | \cdots ; \theta_2)$ are both predicting $x_i$ as the most probable lable, but $p_2$ gives more probabilities to other words (like performing *label smoothing*), PPL will give $p_1(\cdot | \cdots ; \theta_1)$ better performance, even though $p_2(\cdot | \cdots ; \theta_2)$ exhibit better generalizability, costing no performance loss.

Therefore, instead of perplexity, we propose to use *average token rank* to compare across models: for a sentence $x_{1...n}$, for each token $x_i$, given the LLM-predicted conditional probability distribution $\hat{p}(x_i | x_{1...i-1})$, which is a vector $P = (p_1, p_2, ...p_{|\mathcal{V}|})$ where $|\mathcal{V}|$ is the vocab size, we sort the probabilities and get the rank of $x_i$ in it, and average across sentences and datasets. This solves the aforementioned fairness issue to some extent ($p_1(\cdot | \cdots ; \theta_1)$ and $p_2(\cdot | \cdots ; \theta_2)$ are, at least, assigned equal performance scores).

---

[1] https://huggingface.co/spaces/open-llm-leaderboard/open_llm_leaderboard#/

Moreover, since different models have different vocabulary sizes, we also calculate the average token rank proportions, that is, the average token rank of each model divided by its vocabulary size.

Table 6: Average token ranks (and average token rank proportions) on UD-2.9 English/French/Spanish train/dev set

| Language | Model Name | Avg. Rank | Avg. Rank Prop. |
|---|---|---|---|
| English | Meta-Llama-3-8B | 330.48 | 0.26 |
| English | Mistral-7B | 175.10 | 0.54 |
| English | open_llama_7b | 282.89 | 0.88 |
| English | vicuna-7b-v1.5-hf | 1177.39 | 3.68 |
| French | Meta-Llama-3-8B | 275.13 | 0.21 |
| French | Mistral-7B | 212.42 | 0.65 |
| French | open_llama_7b | 123.21 | 0.39 |
| French | vicuna-7b-v1.5-hf | 957.52 | 2.99 |
| Spanish | Meta-Llama-3-8B | 273.54 | 0.21 |
| Spanish | Mistral-7B | 223.06 | 0.68 |
| Spanish | open_llama_7b | 119.66 | 0.37 |
| Spanish | vicuna-7b-v1.5-hf | 919.78 | 2.87 |

## B.2 RESULTS

Results are shown in Table 6. We can see that except for vicuna-7b, other models have similar average token rankings and average token ranking proportions. That might be because vicuna-7b is a instruction fine-tuned model and its attention pattern might be corrupted when not given correct input templates. Among the other three models, open_llama_7b seems to have the best average token rank performances, while Meta-Llama-3-8B is better in terms of token rank proportions. Note that the actual token usage of an LLM given real-world data is actually a long-tail distribution (the most frequently-used tokens constitute the majority of token usage, and most of the tokens are rare ones). Therefore, we take the average token rank as a better criterion of LLMs' "familiarity" of our dataset, and choose open_llama_7b as the representative.

## C MI VISUALIZATION

Our dataset has 45 dependency labels, and open_llama_7b has 1024 attention heads, so it is not feasible to list up all the $MI(\cdot, \cdot)$ values in a table. This means that it's important to choose a good visualization technique, which not only shows as much information as possible, but is also intuitive, easy for us to observe patterns and rules. Therefore, we choose to draw the stack plot of MI values of *all* heads, grouped by layers, with respect to *all* dependency labels. Specifically, for a dependency $l \in \mathcal{L}$, we gather all $\text{MI}_{\text{binary}}(l; A_{b,h})$ values for each head $\langle b, h \rangle \in [1...\mathfrak{b}] \times [1...\mathfrak{h}]$, forming a label MI vector $\mathbf{MI}_l \in \mathbb{R}^{\mathfrak{b} \times \mathfrak{h}}$ We then group $\mathbf{MI}_l$ by layers and draw the stack plot, with the x-axis corresponding to model layer index (let's say $b_0$), and the y-axis corresponding to the sum of all MI values $\sum_{h'} \text{MI}_{\text{binary}}(l; A_{b_0, h'})$. To be noticed, considering the first conclusion we gain in 4.6, if it is a right dependency [2], then we'll choose the $\text{MI}_{\text{binary}}$ values under transposed setting.

In summary, the MI stack plot can: 1. Show the overall trend of MI distribution between layers. 2. Show the MI distribution of heads within a specific layer. 3. By unifying the y-axis, it can also show the MI distribution between different labels. 4. It encapsulates all possible $\langle b, h, l \rangle \in [1 \ldots \mathfrak{b}] \times [1 \ldots \mathfrak{h}] \times \mathcal{L}$.

The MI stack plots are shown in C. Through analysing the visualized MI, we draw the following conclusions:

1. **Label occurrence frequency, tree reconstruction accuracy and MI value are highly related**. Dependencies that are well-captured (like prep,det,nn,pobj,nsubj,amod) also have higher accuracy in

---

[2]The definition is the same as 4.6: if a dependency relation has more arcs where the head is before the dependant, then it is left dependency. On the other hands, right dependencies has more arcs with heads after the dependants

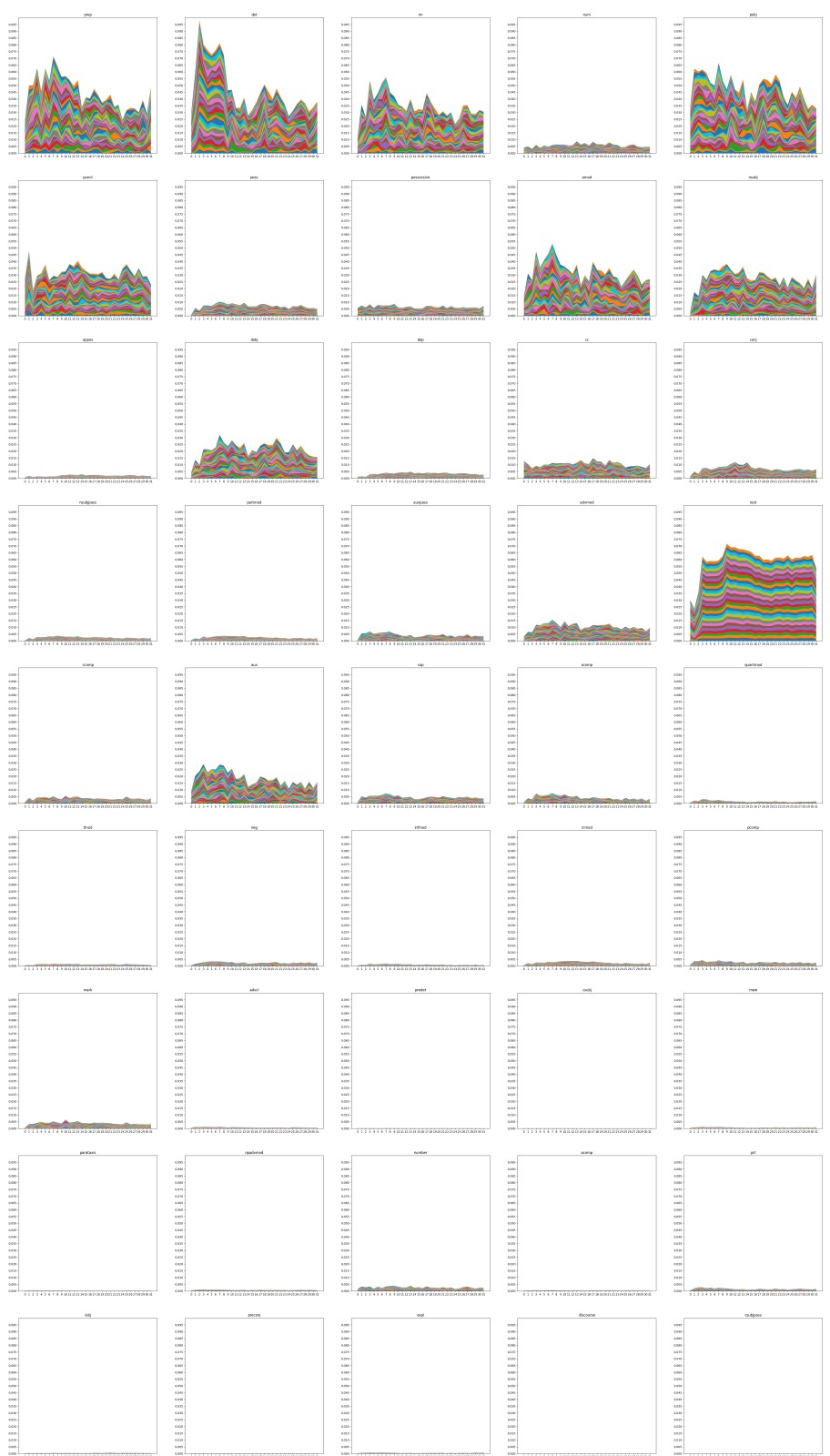

Figure 2: MI stack plot with respect to all labels and heads. The x-axis corresponds to layer index and the y-axis stacks all MI values for all heads within that layer. Different plots corresponds to different dependencies. Depenency root has far more MI compared with other dependencies due to its relative simple pattern, so we divide it by 2 to avoid letting all other dependencies shrink vertically.

reconstructed trees. They're also more frequent in the dataset and real-world language, meaning that dependencies are possibly a emergent ability model learns from data. 2. **Expert heads tend to obey a "principle of locality"**: if a head has high MI at a specific layer, it's plausible that the head's MI is also relatively high in adjacent layers. In the figure, there are many occurrences of this pattern: In layer $b$, head $h$ holds a large portion of MI, and for adjacent layers $b-1, b+1, b-2, b+2, \ldots$, head $h$ also has relatively larger portion of MI. This might indicate that attention Q/K/V or the residual stream *is not rotation invariant*. 3. Among those dependencies that exhibit relatively apparent layer-wise MI differences, lower(local) dependencies have more heads in lower layers, which **further justifies the conclusion model layer correspond to tree layer** in 4.6. A typical example is that dependency det, which is an extremely low-level dependency, has an obvious peak at the first few layers, indicating that the processing of such low-level dependencies might complete within the lower layers, and later layers might just happen to have a similar pattern due to the information passed from the first layers either by residual shortcut or by normal forward pass.

# D  IMPLEMENTATION DETAILS

## D.1  IMPLEMENTATION DETAILS OF IPBP

In this section, we'll briefly introduce the implementation details, like the hyperparameters and key algorithms we use to implement IPBP.

As shown by the source code, we use the PyTorch framework to implement the whole IPBP process. We're not relying on off-the-shelf packages that have KDE functionalities like SciPy[3] and Scikit-Learn[4], since their KDE implementations are CPU-based and thus too in-efficient under our experiment settings.

Specifically, we take samples in $\mathcal{A}_{b,h;l}$ as a whole long tensor $\mathbf{a}_{b,h,l} \in \mathbb{R}^{|\mathcal{A}_{b,h;l}|}$. We calculate the minimum and maximum values of $\mathcal{A}_{b,h;l}$, and build a tensor of real numbers $\mathbf{X} = \{x_1, x_2, \ldots x_{n_x}\}$, ensuring that $x_1 < \min \mathcal{A}_{b,h;l}, x_n > \max \mathcal{A}_{b,h;l}$, and $x_1 < x_2 < \cdots < x_{n_x}$. These discrete $x$ values serve as the points to calculate densities. Next, we calculate the mutual differences between each point in $\mathbf{X}$ and each element in $\mathbf{a}_{b,h,l}$, by repeating $\mathbf{X}^T$ for $|\mathcal{A}_{b,h;l}|$ times, getting a matrix $\overbrace{[\mathbf{X}^T, \ldots \mathbf{X}^T]}^{|\mathcal{A}_{b,h;l}| \text{ times}}$ of shape $n_x \times |\mathcal{A}_{b,h;l}|$, and repeating $\mathbf{a}_{b,h,l}$ for $n_x$ times, also getting a matrix $\overbrace{[\mathbf{a}_{b,h,l}^T, \ldots \mathbf{a}_{b,h,l}^T]}^{n_x \text{ times}}{}^T$ of shape $n_x \times |\mathcal{A}_{b,h;l}|$. The absolute differences of the two matrices $\left| [\mathbf{X}^T, \ldots \mathbf{X}^T] - [\mathbf{a}_{b,h,l}^T, \ldots \mathbf{a}_{b,h,l}^T]^T \right|$, are the mutual differences, let's say $D(\mathbf{X}^T, \mathbf{a}_{b,h,l}) \in \mathbb{R}^{n_x \times |\mathcal{A}_{b,h;l}|}$. Then we calculate the standard deviation of $\mathcal{A}_{b,h;l}$, *i.e.*, $\sigma_{\mathcal{A}_{b,h,l}}$, and take a rule-of-thumb value $\left( \frac{1}{\sum_{i=1}^{|\mathcal{A}_{b,h;l}|} w_i^2} \right)^{-\frac{1}{5}}$ for the bandwidth $B$, with all weight $w_i$s equal to 1. We then calculate element-wise to get the kernel values, following the following equation:

$$\frac{1}{B\sqrt{2\pi \cdot \sigma_{\mathcal{A}_{b,h;l}}}} \exp \left\{ -\left( \frac{D(\mathbf{X}^T, \mathbf{a}_{b,h;l})}{B} \right)^2 \right\} \tag{13}$$

After getting the kernel values in shape $n_x \times |\mathcal{A}_{b,h;l}|$, we calculate the row-wise mean to get the final kernel density values in shape $n_x$. Since all operations of this process are element-wise matrix operations, this is easily parallel-optimizable by PyTorch. As a result, the computation timł for extracting all attention score sets ($\mathcal{A}_{\cdot,\cdot;\cdot}$) and performing all kernel density estimations on the training set is within 1 hour using a single RTX 4090 GPU. Since attention score allocating and distribution estimations are only required to be done once, our method is extremely time-saving compared to most of the supervised probing methods.

---

[3] https://scipy.org/
[4] https://scikit-learn.org/

Table 7: Number of train/dev samples of different language partitions of our dataset

| Language | Split | Number of Samples |
|----------|-------|-------------------|
| English | train | 39832 |
| English | dev | 1700 |
| French | train | 14442 |
| French | dev | 1476 |
| Spanish | train | 14182 |
| Spanish | dev | 1399 |

For inferring on estimated probabilities (like inferring on posteriors $\hat{f}(l|A_{b,h})$ in Section 3.5, we take the estimated posteriors as a set of $n_x$ discrete points, and an attention score in range $[x_i, x_{i+1}]$ will get its corresponding posterior value by interpolating between $x_i$ and $x_{i+1}$. The interpolation-based inferring, together with other processes mentioned in Section 3.5, like head selection and score-weighted averaging, are all parallel-optimized, resulting in being able to run inference within 5 minutes on all baseline settings on a 4090 GPU.

One thing to note is that the Eisner algorithm is originally designed for unlabeled tree extraction, while our method gives probabilities for each label ($P(x_i, x_j; l), \forall l \in \mathcal{L} \cup \{\phi\}$). For each token pair $\langle x_i, x_j \rangle$, we calculate the maximum labeled overall probability $\max_{l \in \mathcal{L}} P(x_i, x_j; l)$ (not including the situation of $l = \phi$) as the arc probability inputs of the Eisner algorithm.

For calculating integrals, specifically, the MI values like $\mathrm{MI}_{\mathrm{binary}}$, $\mathrm{MI}_{\mathrm{pos}}$, we use the trapezoid method to estimate the integral value: as mentioned in the section before, the kernel densities are described by $n_x$ points, we take the $n_x$ points as the integral limits and for every interval between $x_i$ and $x_{i+1}$, we calculate the trapezoid areas and add them up to get the integral values. During tree reconstruction, we empirically set the total number of heads, $i.e.$, $\sum_{l \in \mathcal{L} \cup \{\phi\}} |\mathcal{H}_l|$ to given proportion of total number of heads ($\sum_{l \in \mathcal{L} \cup \{\phi\}} |\mathcal{H}_l| = 2000$ when the model has 1024 heads).

We use the English, French and Spanish partitions of the Universal Dependencies 2.9 (Zeman et al., 2021) dataset. The train/dev sample numbers are shown in Table 7. UD 2.9 is a dependency treebank covering texts from multiple languages and various sources like literature, news articles, spoken languages, *etc.*, with diverse morphological and grammatical features. That dataset is publicly available, using CC BY-SA 4.0 license [5] allowing free redistributions upon notifications. The Universal Dependencies (UD) dataset is designed to provide a standardized framework of grammatical identifications for NLP researchers, so we're following its intended usage.

We applied IPBP on four publicly available LLMs: `open_llama_7b`(Geng & Liu, 2023), `Meta-Llama-3-8B`(AI@Meta, 2024), `vicuna-7b`(Chiang et al., 2023), and `Mistral-7B`(Jiang et al., 2023). We obtained each model's checkpoint from the Hugging Face [6], agreeing on their respective license agreement. Specifically, `open_llama_7b`(Geng & Liu, 2023), and `Mistral-7B` are distributed under the Apache 2.0 license[7]. `Meta-LLamA-3-8B` is released under the Meta LLaMA 3 Community License[8], which allows research use subject to attribution and other conditions. `Vicuna-7B`, which was derived from LLaMA-2(Touvron et al., 2023), follows the LLaMA 2 Community License Agreement[9]. All experiments reported in this paper are conducted strictly within the scope of these licenses' permissions.

### D.2 IMPLEMENTATION DETAILS OF BASELINES

For the $\mathcal{V}$-Information MLPs, we found that training on all datasets will result in a network always predicting $\phi$ for all possible attention scores, due to the long-tail essential discussed in Section 4.4. This will result in many infinite $\mathcal{V}$-Information values, since there will be many estimated probabilities (for label in $\mathcal{L}$ other than $\phi$) rather close to zero. Therefore, we apply a sample balancing technique,

---

[5]`https://creativecommons.org/licenses/by-nc-sa/4.0/deed.en`

[6]`https://huggingface.co/`

[7]`https://www.apache.org/licenses/LICENSE-2.0`

[8]`https://www.llama.com/llama3/license/`

[9]`https://ai.meta.com/llama/license/`

truncating $\mathcal{A}_{\cdot,\cdot;\phi}$ to make their numbers of samples the same as the total number of samples in other score sets $\mathcal{A}_{\cdot,\cdot;1}, \mathcal{A}_{\cdot,\cdot;2}, \ldots \mathcal{A}_{\cdot,\cdot;|\mathcal{L}|}$. What's more, we also did a search on several network sizes, and found that if $\text{MLP}(\cdot)$ is $W_2(\text{act}(W_1(\cdot)))$, where $W_1$ is in shape $1 \times 2$ and $W_2$ is in shape $2 \times 4$ achieves better fitting. This aligns with (Pimentel et al., 2020b) to some extent. We also use PyTorch to implement the baselines. Specifically, for ElasticNet that requires additional training, we use AdamW optimizer, $1e-5$ for both $\lambda_1$ and $\lambda_2$, and use a constant learning rate of $1e-3$, training for 12 epochs. For the $\mathcal{V}$-Information MLP, since we need $\mathfrak{bh}(|\mathcal{L}|+1)$ individual networks for predicting the alternatives of binary MI, we initialize $\mathfrak{bh} \times (|\mathcal{L}|+1)$ sets of matrices, each constituting the weights of a specific network $W_1, W_2, \ldots W_{\text{n\_layers}}$, with $W_1$ having a dimension of 1 and $W_{\text{n\_layers}}$ having a dimension of $|L|+1$. During training and inferring, we concatenate all attention scores $a_{b,h}^{[i][j]}$ for any $b \in \{1 \ldots \mathfrak{b}\}$ and $h \in \{1 \ldots \mathfrak{h}\}$ into a tensor of shape $\mathfrak{bh}$, and use `torch.bmm` to map each element of that tensor to $\mathfrak{bh} \times (|L|+1)$ probabilities (standing for the probabilities of each label conditioned on each attention head's attention score, estimated by the variational family). Using `torch.bmm` will avoid training $\mathfrak{bh} \times (|L|+1)$ networks separately, which is a disaster on computation loads, and can exploit GPU's parallel processing abilities. We use leaky_relu between hidden layers and use sigmoid to form the final probabilities. We use 1e-2 as learning rate with exponential decay (0.8 at each epoch), together with an additional warmup epoch at the beginning. The hyperparameters differ for V-Information since otherwise the variational family network will be more poorly trained. We also trained for 12 epochs.

# E  OTHER STRUCTURAL ABLATIONS

Apart from the formulation of MI and dealing with different dependency directions, we've also done ablations regarding minor design choices.

The first one is for the bandwidth of KDE. Apart from the rule-of-thumb value ($B_0 = \left( \frac{1}{\sum_{i=1}^{|\mathcal{A}_{b,h;l}|} w_i^2} \right)^{-\frac{1}{5}}$), we've also done experiments varying the bandwidth by $0.5 \times B_0$, $2 \times B_0$, and $5 \times B_0$.

Table 8: Ablation results on different bandwidths.

| Bandwidth Multiplier | UAS | LAS |
|:---:|:---:|:---:|
| 0.5× | 49.2 | 30.8 |
| 1× | **49.9** | 34.8 |
| 2× | 49 | 29.8 |
| 5× | 34.8 | 21 |

Results are shown in Table 8. We can see that the rule-of-thumb bandwidth might yield a (locally) optimal result. It can also act as a great influence factor if it is not properly set (such as $5 \times B_0$).

Another ablation is for posterior pooling methods, we proposed to use logarithmic opinion pooling (Heskes, 1997). For comparison, we also used simple arithmetic mean and weighted arithmetic mean. Both of them correspond to the notation in Equation 14, while the former one uses 1 for $w_{\langle b_k, h_k \rangle}$ and the latter one uses $\text{MI}_{\text{binary}}(l; A_{b_k, h_k})$.

$$\text{ArithMean}_{\mathcal{H}_l}(x_i, x_j; l) = \frac{\sum_{\langle b_k, h_k \rangle} w_{\langle b_k, h_k \rangle} \cdot \log \hat{f}(L = l | A_{b_k, h_k})}{\sum_{\langle b_m, h_m \rangle} w_{\langle b_k, h_k \rangle}} \tag{14}$$

Table 9: Ablation results on different posterior pooling methods.

| Pooling Method | UAS | LAS |
|---|---|---|
| Logarithmic Pooling | **49.9** | 34.8 |
| Simple Arithmetic Mean | 36.2 | 18.4 |
| Weighted Arithmetic Mean | 36.3 | 18.9 |

Results are shown in Table 9. We can observe an obvious performance drop when using arithmetic mean methods. We think the reason might be because MI, which characterizes shared bits, vary exponentially (for example, a 2-bit event has $4\times$ numbers of outcomes compared with a 1-bit one), and geometric means naturally captures this. On the other hand, when applying arithmetic means, heads with slightly lower MI (which actually shares much lower information) have similar weights. This may "dilute" and add noise to the pooled probability.

## F  THE USAGE OF LLMS

We used ChatGPT to aid our paper retrieval and literature reviews on fields that we're not familiar with.

