# OpenReview forum: "An Information-Theoretic Parameter-Free Bayesian Framework for Probing Labeled Dependency Trees from Attention Score"
_ICLR.cc/2026/Conference — ICLR 2026 Poster_

### Official Review · Reviewer_7oCt · 2025-10-30

**Soundness:** 2
**Presentation:** 2
**Contribution:** 2
**Rating:** 4
**Confidence:** 4

**Summary:**

The paper aims to reconstruct dependency trees from the attention scores of LLMs using the IPBP method. Precisely, IPBP consists on estimating the Mutual Information between dependencies in a labeled dataset and attention scores. With mutual information, tree reconstruction is performed and evaluated with UAS and LAS metrics. Overall the method proposes a parameter-free way to predict syntax from LLMs. Results show that IPBP is has superior performance to baselines and it works for 7 different LLMs.

**Strengths:**

* The paper spots a limitation of probing techniques and the given solution is simple and elegant.
* The method is well described and mathematically rigorous, both for MI estimation and tree reconstruction.
* The results gather several LLMs and well-thought baselines.

**Weaknesses:**

There are several relevant works which are not cited

* Labeled Dependency probing:
     - https://arxiv.org/abs/2412.05571
* Probing attention values: Attention scores probing:
     - https://www.pnas.org/doi/10.1073/pnas.1907367117
     - https://arxiv.org/abs/1906.02715
* Probing ambiguous sentences with causal analysis:
     - https://arxiv.org/abs/2211.09748
* Syntactic probes are biased towards linearity
     - https://arxiv.org/abs/2508.03211

The results presented fall short at the current version of the manuscript, only different LLMs and baselines are evaluated.
There are no insights about LLM depth and tree depth and how syntactic parsing happens inside the model.

Ablations are based on a single and arbitrary sentence “In order to protect the environment, ecofriendly industries were”, this analysis should be done per dependency type and using specific tokens for the logits and not the average. For example, long-range dependencies like subject-verb agreement in PP sentences are good candidates to assess syntactic capability.

Also, the LAS values are far from SOTA, which might be justified by the fact that it is parameter-free. I think the paper could benefit from a syntactic dataset and thus a more controlled setting.

**Questions:**

How are the multi-token words treated? Are attention values averaged?
How do the maximum/mean layer-wise MI values look like? Within and across dependency types?
What is the baseline MI when using a randomly initialised LLM?

---

> ### Author Response · Authors · 2025-11-19
> **Response to Weaknesses 1&2**
>
> Thank you for your valuable comments! Our reply is as follows:
>
> ### Weakness 1:
>
> First, thank you for reminding us of missing related research. We tried our best to exhaust our knowledge in this field to form a literature review in our paper, but may still miss relevant work. We'll add them to our paper and make it more self-contained.
>
>
> ### Weakness 2:
>
> For the second weakness, actually, we've discussed "model layers correspond to tree layers" in Section 4.5, apologize that the introduction is only briefly mentioned due to page limit.
> Moreover, to take a deeper look inside the dynamic mechanisms of dependency trees' forming during the model's forward pass, we did the following extensive experiment:
>
> We choose several "cut point" layer indices (like 4, 8, ...) and extract dependency trees only using MI and posterior of heads from layers lower than these cut points.
> We think this method somehow shares some ideas with logits lens, since they can both be described as "explaining on partial results to observe tendencies during forward pass". Thus, in our view, it acts as a good simulation for the process of forming emergent dependency trees during the forward pass.
>
> The result is as follows:
>
> |cut point|UAS|LAS|
> |-|-|-|
> |4|24.9|17.8|
> |8|28.4|18.3|
> |12|39.5|23.3|
> |16|43.7|26.1|
> |20|45.8|27.5|
> |24|49.7|30.5|
> |28|49.4|30.3|
> |32|49.2|30.7|
>
> We can see a growing trend of tree quality as the cutting layer becomes higher, and it converges at middle-upper layers. This show that dependency trees emergent along with the forward pass: as the layer become higher, with more attentional computations and heads involved, dependency trees become more complete, and this process finishes at middle-upper layers, with later layers responsible for semantic or logic roles (we though about Tenny et al., 2019 $^{[1]}$ 's work, with yields similar findings on BERT).

---

> ### Author Response · Authors · 2025-11-19
> **Response to Weaknesses 3**
>
> ### Weakness 3:
>
> For the third weakness, we also agree that the ablation discussion should be expanded to cover a wider aspect of syntax.
> A fact that we must agree on is that, if we choose to observe next-token-logits, dependency syntax comprehension might not be directly expressed as logits, since the logits are for _the next token_, but not _this token_.
> Instead, perturbing model's syntax comprehension might affect logits on later words requiring comprehension of some dependencies.
> Therefore, a practical solution is to choose typical syntactical problems and build sufficient templates, then apply the _head-ablation_ techniques on expert heads of relevant dependencies. Specifically, we set the following ablation problems:
>
> - Adjective-modified verb test: This problem builds templates consisting of two confounding adjective-modified subjects and tests whether the model can distinguish the specific subject property.
> It's in the general form of `ADJP[modifier1] NN[subject1] VP[verb1], ADJP[modifier2] NN[subject2] [verb2], Which NN[subject class] [verb1/verb2]? The ADJP1/ADJP2 NN`.
> We ablate the top-100 `amod` expert heads under transposed settings (`amod` is a right dependency) and test logits differences on the token position of the last `JJ1/JJ2`.
> For example, one template is that _The bigger bag is cheaper, and the smaller bag is more expensive. Which bag is cheaper? The bigger bag._, and logits are compared on token _▁bigger_.
>
> - Clausal complement subject finding: This problem tests the model's abilities to comprehend complex nested clauses. It involves templates of more than two-level hierarchical embedded clauses, gives a verb, and tests whether the model knows which embedded clause the verb belongs to. It's in the general form of `NP[main subj] VP[main verb] [NP[clause1 subj\ VP[clause1 verb] [NP[clause2 subj] VP[clause2 verb] ...]]]. WH-word VP[clausei verb]? NP[clause i subj]`. We ablate heads for `nsubj` (transposed) and calculate logits on the last _clause i NP_. An example template is _The teacher believes the student thinks the principal left. Who left? The principal._, and logits will be compared on the last _principal_.
>
> - Indirect object test: In this problem, the model will be given a sentence with a direct object and an indirect object, and will be asked about the indirect object (receiver). It's in the form of `NP[subj] PP[subj confounding modifiers] VP[verb] ADJP[iobj modifier] NP[iobj] ADJP[long confounding modifiers] NN[dobj]. What did NP[subj] VP[verb] NP[dobj] to? The NP[iobj]`. The heads with top-100 `iobj` MI will be ablated, and logits are compared for the answer token `NP[iobj]`. For instance, a template is _The museum curator offered the visiting group a private tour of the restricted halls. What did the curator offer the tour to? The visiting group._, and logits for _visiting_ will be tested.
>
> - Giver/receiver: This also involves an indirect/direct object case, while the model is given two subjects, one will occur later in this sentence, as the giver (subj), and another as the receiver (iobj).
> The model will be asked what the receiver is. The templates are in the form of `[When] NP1 and NP2 VP[wh-clause verb], NP1/2[giver] VP[main verb] NP1/2[receiver, iobj] NP[dobj]`. Top-100 MI heads for `conj` will be ablated, and logits on the receiver NP1/NP2 position will be compared.
> This is actually the problem studied by Wang et al.$^{[4]}$, while they give a circuit based on heuristic experiments. We instead explored this problem in a syntax-intensive view. An example is _When Mary and John went to the store, John gave Mary a drink._.
>
> - Last but not least, long-range subject-verb agreement, which is your suggestion. We build templates in which subjects are modified by a confounding modifier involving similar nouns. The templates are in the form of `NP[subj] PP[confounding modifier] VP[verb] NP[obj]. Who VP[verb]? NP[subj]`. Heads having top-100 MI on nsubj (transposed) are ablated, and logits at the answer `NP[subj]`'s position is tested. A typical example is _The lawyer of the mayor signs documents. Who signed the documents? The lawyer_, which has confounding nouns _lawyer_ and _mayor_.
>
> These problems encapsulate many of the major dependencies and typical problems used in LLM explainability literature. For each problem, 20 templates are constructed, constituting a test set of 100 samples. The logits differences are shown as follows:
>
> |task|avg. logits_diff|
> |-|-|
> |Adjective-modified verb test|-0.51|
> |Clausal complement subject finding|-0.64|
> |Indirect object test|-0.85|
> |Giver/receiver|-1.22|
> |long range subject-verb agreement|-0.57|
>
> We can see that for all problems, tokens' (unsoftmaxed) logits become lower at specific token positions, meaning that on average, model's corresponding dependency comprehending and further generation abilities are hindered by knocking out heads having high MI.

---

> ### Author Response · Authors · 2025-11-19
> **Response to Other Weaknesses and Questions**
>
> ### Weakness 4
>
> For the last weakness, we understand that a syntactical dataset would help. Actually, prior to performing this work, we conducted research on using LLMs to perform supervised dependency parsing on this dataset. The result is that, for a \~7b model and a deep bi-affine neural parser on top of hidden states, if the model is fully tunable, UAS=\~95 and LAS=\~92, if only the parser is tunable, UAS=\~89 and LAS=\~86. Given the fact that 1. The parser operates on hidden states, which consist of more complex semantic/syntactic information compared with attention scores that are purely relational. 2. The tunable parameters and network complexity are significantly higher; we think the UAS/LAS gap might be acceptable. Hope these might help resolve your confusion.
>
> ### Q1
>
> First, apologize that we forgot to introduce treatments for subword-tokenization words, which is important.
> In our implementation, if one work from a dependency arc is subword-tokenized, we use the last subword to stand for it, since decoder LMs use unidirectional attention and information accumulates on later words.
> This is similar to previous syntactical parsing practice on BERT $^{[2][3]}$ that used the first subtoken to stand for the entire word.
> (For detailed code implementation, you may search for `valid_ids`, which is a last subtoken indicator, and `w2s`, which is a mapping to each word's last token index)
>
> ### Q2
>
> For the second question on detailed MI values, we visualized MI using stackplots, which can show MI distribution with respect to _all_ heads and _all_ dependencies intuitively. We've put them in Appendix Section _MI Visualization_, please refer to it for details. Key takeaways from this visualization process include:
>
> 1. Label occurrence frequency, tree reconstruction accuracy, and MI value are highly related.
> 2. Expert heads tend to obey a "principle of locality": if a head has high MI at a specific layer, it's plausible that the head's MI is also relatively high in adjacent layers
> 3. Among those dependencies exhibiting relatively apparent layer-wise MI differences, lower(local) dependencies have more heads in lower layers, which further justifies the conclusion _model layers correspond to tree layers_ in 4.5.
>
> ### Q3
>
> For the last question, we applied the full IPBP process on a transformer model having the same structures as open_llama_7b, with its parameters randomly initialized. We compared the average MI, as well as the reconstructed trees of open_llama_7b and its random counterpart. The results are as follows:
>
> |metric|avg. MI|UAS|LAS|
> |-|-|-|-|
> |open_llama_7b|5.57e-4|49.1|30.6|
> |random (7b)|3.70e-6|12.4|0.76|
>
> We can see that the MI values of the random model are significantly (two orders of magnitude) smaller on average. The trees are also far behind acceptable. This baseline comparison might further indicate that IPBP MI/trees is not a statistical coincidence.
>
> References:
>
> [1] Tenny et al., BERT Rediscovers the Classical NLP Pipeline.
>
> [2] Kitaev and Klein. Constituency Parsing with a Self-Attentive Encoder
>
> [3] Tian et al., 2022. Enhancing Structure-aware Encoder with Extremely Limited Data for Graph-based Dependency Parsing.
>
> [4] Wang et al., Interpretability in the Wild: a Circuit for Indirect Object Identification in GPT-2 small.

---

### Official Review · Reviewer_Y12b · 2025-10-30

**Soundness:** 3
**Presentation:** 3
**Contribution:** 3
**Rating:** 6
**Confidence:** 3

**Summary:**

The paper tackles the problem of probing labelled dependency trees from attention score when a transformers-based LM processes a sentence. Different from several work using deep neural networks (which, according to the paper, is "explaining by unexplainability", the paper proposes to use mutual information between dependency labels and attention scores. Tree reconstruction is done by (i) using the MI to estimate probability that a pair of tokens are linked by a labelled dependency arc, and (ii) using Eisner algorithm.

The paper demonstrates that proposed method can reconstruct (un)labelled dependency tries with higher UAS/LAS scores than the baselines. Interesting, the paper shows that (i) LM decoders capture left/right dependency adaptively, and (ii) different layers capture different types of dependency arcs (local vs global).

**Strengths:**

* The paper is very interesting, especially without any learning. The idea of using mutual information is novel, though using Eisner algorithm in this case is pretty straightforward.

* The real strength of the method lies in its simplicity and explainability.

* Findings in Section 4.5. are particularly interesting, show-casing the strength of the method over learning-based ones.

**Weaknesses:**

* The use of $MI(l,A)$ to compute $P(x_i,x_j;l)$ doesn't seem information-theoretically motivated. How close is it to estimate $P(x_i,x_j;l|A)$?

* It's unclear whether the tree reconstructed by the Eisner algorithm faithfully expresses the syntax structure from the LM. For instance, thanks to the constraints (and biases) of the algorithm, eventhough $P(x_i,x_j;l)$ > $P(x_u,x_v;l)$, the former pair may not form a dependency arc but the latter may. In this counterexample, it's hard to say that the reconstructed tree faithfully expresses the syntax structure of the LLMs.

* The head-selection experiment doesn't capture intrinsic head-selection performance because it is evaluated via the down-stream task (tree reconstruction) using Eisner algorithm. Is there a way to evaluate head-selection performance in an intrinsic manner?

*  The tree-reconstruction evaluation comparison doesn't include results in the literature.

**Questions:**

* How do left/right-branching baselines perform in terms of UAS?

---

> ### Author Response · Authors · 2025-11-24
> **Response to Weakness 1&2**
>
> Thank you for your comments! Our reply is as follows:
>
> ### For Weakness 1
>
> For the first weakness: Yes, performing Logarithm pooling, which requires performing MI-like weighted logarithm mean over MI, may seem information-theoretically confusing. However, it is actually a good probability pooler since it minimizes the total KL-divergence between the overall probability ($P(x_i,x_j,l)$) and each expert's own probabilities ($f(l|a)$), as proved by [1].
>
> Beyond mathematical justification, we also thought about an informal explanation that might foster your understanding: Assume a very special case where all MI values are integers.
> Then, if we regard the pooled outcome as a long 0-1 sequence, the function of MI is clear (also fits its original definition): if head $b,h$ has $m$ bits of MI, then it occupies $m$ bits in the long sequence, and the total sequence length is:
>
> $\sum_{b,h \in \mathcal{H}_l} MI(l;b,h)$.
>
> Then the probability that the overall result is 0 or 1 is:
>
> $\prod_{b,h \in \mathcal{H}_l} {f(l|a)}^{MI(l;b,h)}$
>
> Converting it to a log scale corresponds exactly to logarithm pooling.
>
> We've also compared our logarithm pooling method with other baseline average methods, while simple arithmetic mean (36.2 UAS, 18.4 LAS) and weighted arithmetic mean (36.3 UAS, 18.9 LAS) both caused obvious performance drops. This might add experimental support to the claims above.
>
> ### For Weakness 2
>
> We understand that the trees reconstructed by Eisner might not correctly reflect the information trend for dependency embedded in model attention, since Eisner is based on DP and globally optimized. We designed an alternative IPBP process using the maximum spanning tree algorithm, which is more greedy (and thus more favorable for higher $P(x_i,x_j;l)$), for tree reconstruction.
> We noticed that, since MST is undirected and decoder LLM's causal attention is also unidirectional, they're creating a good fit.
> So during attention score collecting and density estimation, we correspond the upper triangular area of tree adjacency matrix to the lower triangular area, also creating a "causal" (and can be seen as undirected) tree adjacency matrix, and collect $\mathcal{A}_{\cdot,\cdot;\cdot}$ sets and estimate densities according to that matrix. During tree reconstruction, we use Kruskal's maximum spanning tree algorithm to get dependency trees based on $P(x_i,x_j;l)$. We evaluate the undirected counterpart of the tree metrics (UUAS and ULAS). The result is as follows:
>
> |UUAS|ULAS|
> |-|-|
> |45.3|28.4|
>
> We can see that, the spanning dependency trees are of similar quality compared with those decoded by Eisner. Moreover, converting the dependency adjacency matrix to "causal" ones actually corresponds to the process of "decoder models adaptively capture left/right dependencies", which is the first claim we've made in Section 4.5, so the results actually justified that claim from another perspective.

---

> ### Author Response · Authors · 2025-11-24
> **Response to Other Weaknessed & Questions**
>
> ### For weakness 3
>
> Yes, we also agree that evaluating a feature selection process cannot be verified solely by downstream task performance. To solve this, we adapted Fan et al $^{[2]}$'s intrinsic evaluation framework, which only relies on the feature selection (in our case, head selection) results to evaluate themselves. Their method is based on the voting theory, which assumes that a good selection method will also be favored by (thus being more consistent with) other baselines participating in the comparison. By the way, we think that this is actually a little bit like conferences nowadays that let authors also serve as reviewers.
>
> Specifically, to adapt Fan's method, a criterion indicating _how similar two head-selection methods are_ is required. Since each head selection method (and IPBP) yields a correlationship matrix in shape $[|\mathcal{L}|, \mathfrak{b}\times\mathfrak{h}]$ (with respect to each dependency-attention head compositions), for each $l\in\mathcal{L}$, we calculate a Spearman correlation between the corresponding rows of two head-selection methods, and average them to get the macro-Spearman corr. of two head-selection methods. The corresponding macro-Spearman matrix is as follows:
>
> | |Probless|IoU|ElasticNet|IPBP|Row Average (without diagonal)|
> |-|-|-|-|-|-|
> |__Probless__|1.000|0.198|0.276|0.327|__0.267__|
> |__IoU__|0.197|1.000|0.224|0.438|__0.286__|
> |__ElasticNet__|0.276|0.224|1.000|0.427|__0.309__|
> |__IPBP__|0.327|0.438|0.427|1.000|__0.398__|
>
> We can see that IPBP shares the most rankings with other baselines, not only individually compared but also on average, thus it has the best intrinsic performance under Fan's evaluation framework.
>
> ### For Weakness 4
>
> Yes, previous probing methods (a typical example is [3]) are not evaluated.
> Like as shown in Section 2, some of them probe on attention scores, while others focus on hidden states. We probe on attention scores. For attention score-based probing baselines, actually, the Raw Score baseline corresponds to the methodology of a lot of previous attentional probing methods $^{[4][5][6]}$. For those focusing on hidden states, since 1. hidden vectors contain much richer representations compared with attention scores (empirical support is that hidden vectors have much larger dimensionalities). 2. They involve training additional networks, usually a deep and complex one ([7][8]). Given the fact that one of our goals is to ensure a transparent method that avoids the _explain by unexplainability_ issue as stated in Section 1, we think they may act as a ceiling indicating how much we can extract from all possible components in LLMs, instead of baselines to compare. Again, sorry for the confusion, and we'll state the above purpose more clearly in Introduction/Experiments.
>
> ### For Questions
>
> Simple branching can act as a good and easy-to-implement parameter-free baseline, but we forgot this. Thank you for reminding!
> We've done the experiments, and the results are as follows:
>
> |Branching|UAS|
> |-|-|
> |Left|16.6|
> |Right|26.4|
>
> We can see that right-branching performs better, with a UAS of 26.4, but is still lower than the worst-performing baseline (Raw score, 32.3).
>
> If you have further questions and concerns, feel free to discuss!
>
> References:
>
> [1] Ali E. Abbas. 2009. A Kullback-Leibler View of Linear and Log-Linear Pools.
>
> [2] Fan et al., 2024. Evaluating Neuron Interpretation Methods of NLP Models
>
> [3] Manning et al., 2019. A Structural Probe for Finding Syntax in Word Representations.
>
> [4] Clark et al., 2019. What Does BERT Look at? An Analysis of BERT's Attention.
>
> [5] Vig and Belinkov, 2019. Analyzing the Structure of Attention in a Transformer Language Model.
>
> [6] Ravishankar et al., 2021. Attention Can Reflect Syntactic Structure (If You Let It).
>
> [7] Pimentel et al., 2020. Information-Theoretic Probing for Linguistic Structure.
>
> [8] Pimentel et al., 2022. Attentional Probe: Estimating a Module’s Functional Potential.

---

> > ### Comment · Reviewer_Y12b · 2025-11-26
> >
> > I would like to thank the authors for the response.
> >
> > Re weakness 2, I would like to clarify what I meant by "faithfully": when we use any dependency parsing algorithms (e.g. Eisner, MST), we effectively bias our interpretation about syntactic structures given by LLMs towards our expectation. For instance, it could be that LLMs allow a dependent to have two or more heads even though we expect only one head.

---

> ### Author Response · Authors · 2025-11-27
> **Further Replies on Weakness 2**
>
> Thank you for further clarifying weakness 2! We now understand what you mean by "faithfully": it means that the underlying emergent linguistic structures learned by LLMs do not have to be dependency trees, but probing assumes it to be.
> Actually, it is a problem with respect to probing (also including all other feature attribution-based interpretability methods) itself, not specific to our method. Based on our understanding, we think assuming a human-defined concept in prior is a "necessary evil" for two reasons:
>
> First, models' internal states are usually noisy and superpositioned (See [1]), meaning that assuming human-understandable concepts in prior and extracting them from model states for our convenience might be inevitable. In order to verify this, we think an entropy comparison experiment between $A_{\cdot,\cdot}$ (the resulting density is $f(a)$) and $L$ might be helpful. However, $A_{\cdot,\cdot}$ is continuous and $L$ is discrete, while it's an open problem to estimate continuous entropies, with analogs such as LDDP proven to be non-positive, also not sharing the scale with discrete Shannon entropies. To solve these two problems, we first bucket $A_{\cdot,\cdot}$ to form histogram discrete distributions $P(A\in A_i), i={1...N}$. Together with the Bernoulli distribution $P(L)$, we calculate the KL-divergences with respect to discrete uniform distributions having the same number of values. (For example, for $P(A)$ having $N$ possible values, we calculate $\sum_{i=1}^N {P(A\in A_i) \log{[N\cdot P(A\in A_i)]}}$. This is principally similar to LDDP.) Given that this measure also characterizes the number of extra bits to code ($P(A)$ or $P(L)$) using a uniform code, we divide it by the uniform code length ($\log{N}$ or $\log{2}$) for a uniform scale. This "normalized KL-divergence" characterizes "distribution tidiness": the higher the value is, the more distant the distribution is from a uniform discrete distribution with the same cardinality. On the contrary, it might be more evenly distributed, containing more chaos. Results of the (normalized) KL-divergences averaged across all heads/all dependency labels are as follows. We can see that attention is much noisier compared with human-friendly concepts like dependencies. This entails a feature extraction method (like probing), including setting an inductive bias on concepts to extract.
> |Distribution|$P(L)$|$P(A)$|
> |-|-|-|
> |Avg. Normalized KLD|0.99|0.67|
>
> Second, while models' internal mechanisms might not perfectly correspond to those prior concepts, important subforms of them are usually reflected. Htut et al$ ^{[2]}$ once verified that, for an attention head, even though they simply picked the most attended tokens, i.e., tokens with the largest attention weights. This is a simple unbiased strategy (they said "the relations extracted using this method need not form a valid tree, or even be fully connected, and the resulting edge directions may or may not match the canonical directions."). They find some heads who dedicate a considerable amount of the most attended tokens to some dependency relationships. We replicated their experiments on open_llama_7b attention scores and results (top20) are as follows:
>
> |Dependency|Acc. by Maximum Weight|
> |-|-|
> |root|100.0|
> |csubjpass|100.0|
> |iobj|95.24|
> |expl|93.94|
> |possessive|92.63|
> |prt|92.24|
> |mwe|91.07|
> |auxpass|86.12|
> |pobj|85.28|
> |det|82.94|
> |dobj|74.69|
> |preconj|72.73|
> |aux|68.62|
> |infmod|67.78|
> |amod|67.26|
> |acomp|66.67|
> |neg|66.27|
> |npadvmod|65.17|
> |pcomp|64.11|
> |number|62.68|
> |The Rest(average)|39.32|
>
> We're surprised to observe that, a great many dependencies (especially those frequently-occurring dependencies) can be discovered by simply picking the token with maximum weights! (This result is higher than Htut's results on BERT, indicating that LLMs might be more syntactically-intensive due to more heads). This means that even though perfect dependency trees might not be implicitly formed, lots of dependency relationships are still well-captured by model attentions, and setting a dependency tree prior is an acceptable reduction.
>
> We think these two reasons might explain why lots of previous researchers take this inductive bias as acquiescence and have developed corresponding methods. We hope this helps resolve your confusion. If you have further questions, please feel free to discuss!
>
> References:
>
> [1] Elhage et al., 2022. Toy Models for Superpositions.
>
> [2] Htut et al., 2019. Do Attention Heads in BERT Track Syntactic Dependencies?

---

### Official Review · Reviewer_adce · 2025-10-31

**Soundness:** 2
**Presentation:** 2
**Contribution:** 2
**Rating:** 4
**Confidence:** 2

**Summary:**

This paper proposes a parameter-free probing method for detecting how well attention maps in decoder-only transformers encode dependency tree information. This approach relies on estimating the mutual information between the attention scores of a pair of tokens and the dependency label, and no added learning of probe weights. This approach is evaluated for a number of models and compared against various baselines on UAS and LAS. Results show that this approach is revealing more syntactic structure than previous methods.

**Strengths:**

Parameter-free probes generally seem like a good idea and this approach generates MI estimates that can be useful for more fine-grained analysis.
The paper is thoughtful about considering a number of different baselines in their experiments.

**Weaknesses:**

There is a mismatch between the attention scores being used for this analysis (bidirectional) and the actual scores being used by the model during generation (causal). It seems a bit strange to make claims about the syntax structure encoded in the model that is never actually used for generation. This point makes the conclusions and implications of the findings a bit hard to reason about.

**Questions:**

-

---

> ### Author Response · Authors · 2025-11-16
> **Experiments Probing Causal Attention Instead of Reconstructed Bidirectional Attention**
>
> Thank you for your valuable comments! Our reply is as follows:
>
> We agree that reconstructing bidirectional attention scores may bring discrepancies, and you're right to notice that. We're also curious about whether or not this can be avoided. Through analysing, we found that actually using Q/K-reconstructed attention scores is _a compromise to directed dependency parsing_, since it is theoretically impossible to map the directed dependency adjacency matrix to the lower triangular area and reconstruct without losses. On the other hand, this also means that if we _relax the directed tree's restriction_, relying on "useless" scores can be avoided (actually lots of previous work probe for undirected trees $^{[1][2][3]}$, so we think it's ok to relax that). Specifically, we tweak our methods as the following to perform undirected MI estimation and tree extraction:
>
> When gathering attention scores (corresponding to Section 3.2), for right dependencies, we allocate the attention score at their transposed position in attention matrices, as follows:
>
> $\\mathcal{A}_{b,h;l^{[i][j]}}  \\cup=  $
>
> $\\begin{cases} a_{b,h;l^{[i][j]}}, &\\text{if} i > j \\\\ a_{b,h;l^{[i][j]}}, &\\text{else}\\\\ \\end{cases} (1) $
>
> It can also be considered as "folding" (transposing) the upper triangular part of the original dependency adjacency matrix to the lower triangular part.
>
> During decoding, Eisner algorithm is no longer applicable for undirected trees, so we instead use the maximum spanning tree algorithm to decode the tree. Specifically, after we get the overall probabilities $P(x_i, x_j, l)$ (as in Section 3.5), we take it as the weights of edges and apply Kruskal's algorithm to form a spanning tree. We calculate the undirected unlabeled/labeled attachment scores (UUAS/ULAS), similar to the UAS/LAS scores we used for directed trees, with the treebank's ground-truth trees. Results on open_llama_7b is as follows:
>
> |UUAS|ULAS|
> |-|-|
> |45.3|28.4|
>
> The resulting tree is with similar qualities as the directed trees from bidirectional scores recorded in our paper, approximately the average value of the results for original/transposed settings. This indicates that it is feasible to extract undirected dependency trees using IPBP method, based on the down-triangular (not masked during generation) attention scores. Moreover, the results also further justifies our assumptions in Section 4.5 (_decoder models adaptively capture left/right dependencies_), since eq(1) is exactly letting attention scores adaptively stand for left/right dependencies, and the result equals to the average.
>
> Thank you for reading. If you have further questions, feel free to discuss!
>
>
> References:
>
> [1] Hewitt and Manning, 2019. A Structural Probe for Finding Syntax in Word Representations.
> [2] Manning et al., 2020. Emergent linguistic structure in artificial neural networks trained by self-supervision
> [3] White et al., 2021. A Non-Linear Structural Probe.

---

### Official Review · Reviewer_nQmi · 2025-11-02

**Soundness:** 3
**Presentation:** 3
**Contribution:** 3
**Rating:** 6
**Confidence:** 2

**Summary:**

This paper takes a step to address how LLMs comprehend syntax by proposing a novel method to extract labeled dependency trees directly from attention scores without training any additional neural networks. They aim to solve two problems: Directed Tree Extraction and Mutual Information Estimation for LLMs in a parameter-free Bayesian framework. Experiments are conducted comparing against several baselines, including probing and V-information comparing on labeled attachment scores (LAS), and unlabeled attachment scores (UAS) , where they show superiority of their method.

**Strengths:**

I found the paper overall easy to read, and the proposed Bayesian approach is sound, and I think the baselines considered are sufficient.

**Weaknesses:**

I am not an expert in this specific area, so my comments focus on high-level suggestions that could help improve the paper overall.

1. How does this work compare with Wu et al. (2020)’s *Perturbed Masking*, which is mentioned as a parameter-free approach? It would be valuable to include a direct comparison on the same datasets and metrics to better position the proposed method.

2. Could the authors consider a simpler baseline by using averaged attention scores directly as edge weights for Eisner or Chu-Liu-Edmonds decoding? This would help clarify how much improvement comes from the proposed components.

3. For ablations, it would be helpful to vary the bandwidth parameter for KDE across a range (for example, 0.5x, 1x, 2x, and 5x of the current setting) and report UAS and LAS results to evaluate sensitivity.

4. Another useful ablation would be to compare logarithmic opinion pooling with simpler aggregation methods such as the arithmetic mean or the standard geometric mean to understand whether the more complex approach provides a meaningful benefit.

5. The motivation and implications section could be made stronger. At present, the introduction seems to overstate the practical and scientific significance given the modest LAS scores of around 30 to 49\%. It would help if the authors discussed more candidly when attention-based syntax extraction is sufficient, when hidden states are preferable, and what these results reveal about whether large language models actually encode syntactic structure.

**Questions:**

Refer to the Weakness

---

> ### Author Response · Authors · 2025-11-24
> **Response to Weakness 1-3**
>
> Thank you for your comments! Our reply is as follows:
>
> ### Weakness 1
>
> Yes, Perturbed Masking is a good unlabeled probing baseline, thank you for proposing this! Even though it is originally only applicable to MLMs like BERT, and it operates on hidden states instead of attention scores, we're also curious about its performance on decoder LLMs, so we decided to adapt it to the open_llama_7b model and do experiments on our dataset.
>
> However, applying it on decoder models is not easy, since: 1. Decoder models usually don't have `[MASK]` tokens since their unsupervised objective is usually language modeling. 2. Decoder models are unidirectional, so perturbing one token cannot affect previous tokens. 3. Decoder models' last hidden state is responsible for _next token generation_ instead of _this token's mask prediction_. Therefore, some questions naturally appeared, like 1. How to mask and perturbate? and 2. For token $x_i$, which last hidden state should we select as its influence evidence?
>
> We've considered the following solutions for the above questions:
>
> For masking/perturbing method:
>
> 1. Replacing the token embeddings of masked tokens with the average token embeddings of all tokens in the vocabulary.
>
> 2. Replacing the masked token with a common word (like `▁the`)
>
> 3. Replacing the masked token with a random token, sampled from the token distribution on our dataset.
>
> For hidden state selection, for token $x_i$, we considered using its last hidden state for the influence evidence of $x_i$ itself and $x_{i+1}$ (shift right).
>
> Moreover, for the criterion of "influence on other tokens after masking", like the original paper, we selected the Euclidean distance and the logits difference.
>
> Finally, due to the aforementioned unidirectionality limitation, the influence matrix must be upper triangular since masking one token only affects later tokens, so we choose the spanning tree decoding algorithm and get undirected trees.
>
> We did experiments for different compositions of these different alternative solutions on a small validation set (consisting of 100 samples). Some representative results are as follows:
>
> |Mask Method|Hidden State Selection|Influence Cirterion|UUAS|
> |-|-|-|-|
> |Mean embmdding mask|shift right|Euc. dist.|18.2|
> |Fixed token(▁the) mask|shift right|Euc. dist.|36.5|
> |Random token|shift right|Euc. dist.|39.4|
> |Random token|shift right|Logits diff.|28.6|
> |Random token|no shift right|Euc. dist.|30.6|
>
> By comparing as above, we found that _Random token+shift right+Euclidean distance_ is the best practice so far. By applying this composition on the full validation set, the result is 38.7 UAS, slightly lower than a fixed branching baseline (41.0). Our experiment results indicate that, when applied to decoder models, perturbed masking might not be as effective as its original MLM-based use.
>
> ### Weakness 2
>
> Actually, the _Raw Score_ tree reconstructed baseline is exactly the simple baseline using averaged attention scores. Sorry for the confusion!
>
> ### Weakness 3
>
> We've done ablations under 0.5x, 2x, and 5x KDE bandwidth settings. The results are as follows:
>
> |Bandwidth Multiplier|UAS|LAS|
> |-|-|-|
> |0.5x|49.2|30.8|
> |2x|49|29.8|
> |5x|34.8|21.1|
>
> Results indicate that the rule-of-thumb bandwidth might yield a (locally) optimal result. It can also act as a great influence factor if it is not properly set (5x), so doing these ablations is necessary to make our paper more rigorous. Thank you for reminding!

---

> > ### Author Response · Authors · 2025-11-24
> > **Response to Weakness 4&5**
> >
> > ### Weakness 4
> >
> > We've compared the logarithmic pooling method (which itself is a geometric mean) with simple (equal weights) arithmetic mean and MI-weighted arithmetic mean. Tree reconstruction results are as follows:
> >
> > |Average Method|UAS|LAS|
> > |-|-|-|
> > |Logarithmic (ours)|49.9|34.8|
> > |Simple Arithmetic Mean|36.2|18.4|
> > |Weighted Arithmetic Mean|36.3|18.9|
> >
> > We can observe an obvious performance drop when using arithmetic mean methods. We think the reason might be because MI, which characterizes shared bits, vary exponentially (for example, a 2-bit event has 4x outcomes compared with a 1-bit one), and geometric means naturally captures this. On the other hand, when applying arithmetic means, heads with slightly lower MI (which actually shares much lower information) have similar weights. This may "dilute" and add noise to the pooled probability $P(x_i, x_j, l)$.
> >
> > ### Weakness 5
> >
> > Yes! We'll make modifications to the Introduction section and make it more experiment-grounded. During this author-reviewer discussion period, to answer the questions from all reviewers, we've done experiments about how extracted dependency trees evolve during the forward pass, how expert heads influenced next token generation through a mechanism perspective, and IPBP performance under many alternative settings. These might reveal a lot about LLMs' emergent syntax structure. What's more, these results also give clues about the pros and cons of attentional probing. Together with some existing discussions ([1][2][3][4]), we'll make the comparisons between these two probing more objective and rigorous.
> >
> > Again, thank you for your valuable comments, and please feel free to discuss!
> >
> > References:
> >
> > [1] Hewitt and Liang, 2019. Designing and Interpreting Probes with Control Tasks.
> >
> > [2] Pimentel et al., 2022. The Architectural Bottleneck Principle.
> >
> > [3] Jain and Wallace, 2019. Attention is not Explanation.
> >
> > [4] Wiegreffe and Pinter, 2019. Attention is not not Explanation.

---

> > > ### Comment · Reviewer_nQmi · 2025-11-24
> > > **Thanks for addressing the questions.**
> > >
> > > I don't have any other questions; I will retain my positive scores.

---

### Author Response · Authors · 2025-12-03
**Overview of Reviewer Feedback and our Resolutions**

Dear AC/SAC:

To help reduce your workload, we've made a brief summary of the addressed/leftover issues during this discussion:

Reviewer __nQmi(6)__ admitted and appreciated our response. The confusions are resolved, and we've done experiments covering all his/her suggestions.

Reviewer __Y12b(6)__ appreciated our earlier response, and further clarified on what he/she meant by W2 (the bias introduced by dependency prior). We replied with mathematical theoretical support (calculating uniform codelength-normalized KL-divergence), experimental validations (expert heads still excel in dependency labeling under unbiased maximum-weight prior), and also noted that assuming dependency trees is common practice for the probing task. It's a follow-up question, so the response was posted on 28th Nov. The reviewer didn't have enough time to reply then.

Reviewer __adce(4)__'s only stated weakness is the mismatch between our reconstructed attention and causal attention. We designed the MST-based IPBP fully conditioning on causal attention, which resolved this mismatch and got equivalent results, directly resolving the concern.

For reviewer __7oCt(4)__, we conducted experiments including: 1. progressively evaluating tree-reconstruction layerwise, 2. a comprehensive ablation involving five tasks and a 100-sample dataset, 3. results on parsing instead of probing, 4. MI stackplot visualization and interpretation, and 5. baseline MI of LLMs with random weights. We also clarified the first conclusion in Section 4.5 and our treatments for subword tokenization to resolve his/her other misunderstandings.

While reviewers adce and 7oCt haven't replied yet (likely due to time issues), we still think the above experiments/explanations completely addressed their proposed weaknesses/questions. Some of these additional experiments (like the syntactical ablation tasks and dataset for logits difference) might even spur individual research upon further exploration. Please refer to our response for details.

Again, we sincerely appreciate your time and efforts, and thank you for reading!

---

### Meta-Review · Area_Chair_UVKX · 2026-01-04

**Summary:**

## Summary

The paper proposed a novel method to extract labeled dependency trees directly from attention scores without training any additional neural networks to address how LLMs comprehend syntax.

The core idea is to treat dependency labels as discrete variables and attention scores as continuous variables, then use KDE-based distribution estimation to compute (binary) mutual information between each head and each dependency relation, and further derive Bayesian posteriors for arc labeling.



## Overall Score

nQmi: 6 (maintain positive score)

adce:  2 (no response)

Y12b: 6 (maintain positive score)

7oCt: 4 (no response)

## Addressed Concerns

* Further details of the ablation  and motivation (nQmi, Y12b,70Ct)
* Mismatch between the attention scores being used for this analysis and the actual scores (adce)
* Explanation of the Equation in method (Y12b)
* Missing references (7oCt)
* Missing some specific datasets, like syntactic datasets  (7oCt)
* Further explanation of some definition (nQmi, 7oCt)

## Remaining Controversial Concerns

* Further discussion on whether the tree reconstructed by the Eisner algorithm faithfully expresses the syntax structure from the LM (Y12b)
* More comprehensive experiments and ablation studies would be helpful (nQmi, Y12b, 70Ct)

## Conclusion:

Overall, during the discussion phase, two reviewers (nQmi and Y12b) were satisfied with the authors' responses and kept their positive scores. As for the concerns raised by Reviewer 7oCt and Reviewer adce, the authors provided detailed and convincing responses. That said, neither of these two reviewers followed up afterward.

During discussion, the authors did a solid job addressing the reviewers' questions and concerns. While there are still some minor issues with experimental details and notation, these can be refined in a revision.

In short, I recommend accepting this paper.

**Reviewer Concerns:**

Refer to Summary

**Reviewer Scores:**

Refer to Summary

---

### Decision · Program_Chairs · 2026-01-26

Accept (Poster)